# Pervasive phenotypic effects of FBXO42 are promoted by regulation of PP4 phosphatase

Hongbin Yang[1,8], Paul Smith [ID][2,8], Yingying Ma[3], Emily Southworth [ID][2], Varun Gopala Krishna[2], Beatrice Salerno[2], Joseph Rowland[2], Alexander E P Loftus[2], Domenico Grieco[4,5], Iolanda Vendrell[6,7], Roman Fischer [ID][6,7], Benedikt M Kessler [ID][6,7] & Vincenzo D'Angiolella [ID][2✉]

## Abstract

F-box proteins are the substrate recognition modules of the SCF (SKP1–Cullin–F-box) E3 ubiquitin ligase complex. FBXO42, an understudied member of this family, has recently emerged as a modulator of key cellular processes, including cell cycle progression, the DNA damage response, and glioma stem cell survival. In this study, we define the function of FBXO42 as a major regulator of the protein phosphatase PP4. Phosphoprotein phosphatases (PPPs) have a broad array of substrates, hence necessitating tight regulation. We observe that FBXO42 ubiquitinates the PP4 complex to govern the assembly of regulatory and catalytic subunits, with the net effect of restraining the latter's phosphatase activity. FBXO42 depletion unleashes PP4 activity, with broad cellular effects, highlighting FBXO42 as a novel regulatory node in ubiquitin-mediated signalling for future therapeutic exploitation.

**Keywords** Ubiquitin; F-box Proteins; FBXO42; PP4 Phosphatase; CRLs
**Subject Category** Post-translational Modifications & Proteolysis

## Introduction

Modification by ubiquitin is a widespread signalling cascade to regulate essential biological processes (Damgaard, 2021). The system operates by the sequential addition of ubiquitin moieties to form chains on substrates, which are recognized by the proteasome for degradation (Hershko and Ciechanover, 1992).

The fate of ubiquitinated proteins depends on a code largely reliant on polyubiquitin chain topologies thus generated (Komander and Rape, 2012). Ubiquitin moieties linked through lysine 48 (Lys48) or 11 (Lys11) are typically degradative signals, while polyubiquitin chains harbouring alternate 'atypical' linkages have other distinct outcomes. In many instances, non-canonical polyubiquitin chains serve as scaffolds to recruit protein complexes (Agrata and Komander, 2025).

The specificity in the ubiquitin system is dictated by a hierarchical system of three components: ubiquitin activating (E1), conjugating E2 and ligase (E3s) enzymes. E3s are comprised of large families of enzymes which select substrates based on specific assembly modules (Morreale and Walden, 2016). Most abundant among the E3s, Cullin Ring Ligases (CRLs) use a central component (cullin) to bridge a substrate recruitment protein and a RING Finger containing protein, which, in turn, recruits E2 enzymes for ubiquitin transfer (Harper and Schulman, 2021; Zheng et al, 2002). F-box proteins are the substrate adaptor modules of cullin 1 (Cul1) complexes, the module being composed of S-phase kinase-associated protein 1 (Skp1), Cul1, and the F-box itself (SCFs). These are further sub-classified as F-boxes with a WD-40 domain (FBXWs), F-boxes with a leucine zipper domain (FBXLs) and F-boxes with other domain (FBXOs) based on their substrate recruitment domains (Cardozo and Pagano, 2004; Fouad et al, 2019).

Prototypical F-box proteins engage substrates by binding to discrete sequences called "degrons". As an example, Cyclin F (FBXO1) recognizes a bivalent domain (F-deg) composed of a phosphorylated residue and a cyclin binding domain (RxIF) (D'Angiolella et al, 2012; Mavrommati et al, 2018; Ngoi et al, 2025; Yang et al, 2024). Most of the substrates are recognised in G2/M when the level of cyclin F itself peaks, allowing the control of multiple substrates by a single E3. As exemplified above, there is significant interplay between phosphorylation and degradation, with some F-boxes (particularly FBXWs) recognising substrates only after phosphorylation (Jin et al, 2005).

Phosphorylation is a reversible modification balanced by the opposing action of kinases and phosphatases. Phosphoprotein phosphatases (PPPs, PP1-PP7) use a common catalytic process which entails activation of a water molecule acting as the nucleophile in the dephosphorylation reaction (Shi, 2009). Albeit similar in mechanism of action, PPPs differ significantly in subunit composition and assembly. PP4 uses a total of five regulatory subunits to assemble distinct hetero-oligomeric complexes. Indeed, PP4C (the catalytic subunit) can assemble with PP4R1 or PP4R2,

[1]Lee Kong Chian School of Medicine, Nanyang Technological University, 11 Mandalay Road, Singapore 308232, Singapore. [2]Edinburgh Cancer Research, CRUK Scotland Centre, Institute of Genetics and Cancer, University of Edinburgh, Crewe Road South, Edinburgh EH4 2XU, UK. [3]Tsinghua University, Beijing, China. [4]DMMBM, University of Naples "Federico II", via S. Pansini 5, 80131 Naples, Italy. [5]CEINGE Biotecnologie Avanzate "Franco Salvatore", via Gaetano Salvatore 486, 80145 Naples, Italy. [6]Target Discovery Institute, Centre for Medicines Discovery, Nuffield Department of Medicine, University of Oxford, Oxford OX3 7FZ, UK. [7]Chinese Academy for Medical Sciences Oxford Institute, Nuffield Department of Medicine, University of Oxford, Roosevelt Drive, Oxford OX3 7FZ, UK. [8]These authors contributed equally: Hongbin Yang, Paul Smith. ✉E-mail: vdangio@ed.ac.uk

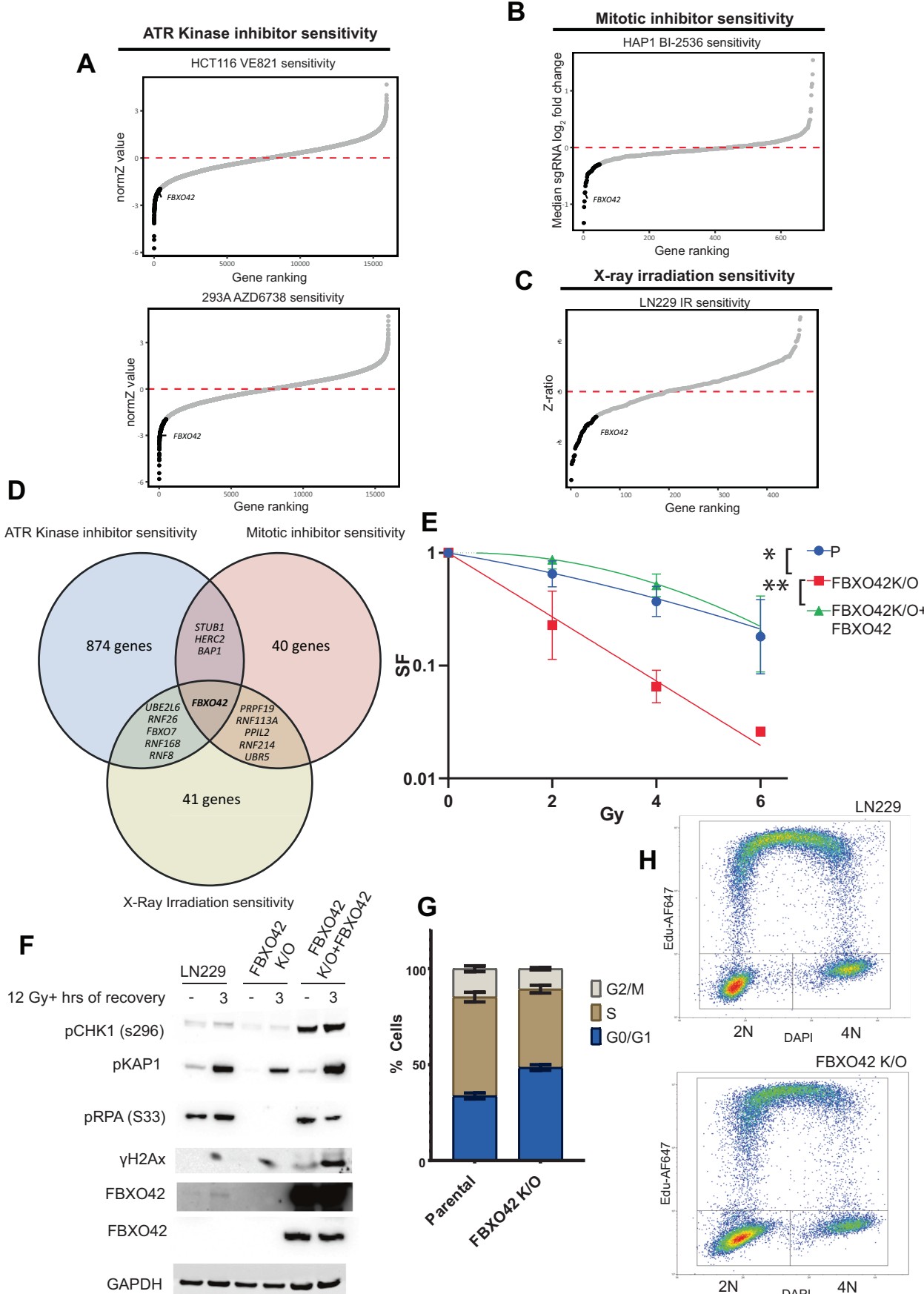

**Figure 1. Loss of FBXO42 induces multiple cellular defects including defective DDR responses.**

(A) CRISPR screen probing vulnerabilities in response to inhibitors of the ATR kinase from (Hustedt et al, 2019; Wang et al, 2019) in HCT116 and 293A cells. Outcome of screens are represented by gene ranking using Z-Score (normZ) values, using a significance cut-off of ≤−1.96. FBXO42 is highlighted. (B) CRISPR screen probing vulnerabilities in response to inhibitor of the kinase PLK1 from (Hundley et al, 2021) in HAP1 cells. Outcome of screen is represented by gene ranking using median sgRNA log2 fold change cut-off of ≤−0.3, FBXO42 is highlighted. (C) CRISPR screen probing vulnerabilities in response to IR (Yang et al, 2024) in LN229 cells. Outcome of screen is represented by gene ranking using Z-ratio values, using a significance cut-off of ≤−1, to reflect the use of a focussed CRISPR-Cas9 screen in place of genome-wide. FBXO42 is highlighted. (D) Venn diagram of hit genes identified in the CRISPR screens depicted in (A–C), displaying hits unique to each screen and those common across 2 or more screens. (E) Colony formation assay in FBXO42 K/O cells. LN229 parental cells, LN229 FBXO42 K/O cells or FBXO42 K/O cells where FBXO42 was re-expressed under the control of a doxycycline inducible promoter were seeded for colony formation assay and challenged with the indicated dose of IR. Seven days after IR, cells were stained with crystal violet and counted. Data presented as mean +/− SD; n = 3 biological replicates. Two-tailed unpaired t test was performed as statistical analysis. Exact P values for P vs FBXO42 K/O is 0.03 and FBXO42 K/O vs FBXO42 K/O + FBXO42 is 0.01. *P ≤ 0.05, **P ≤ 0.01; ***P ≤ 0.001; ****P ≤ 0.0001. (F) Immunoblotting after IR treatment and 3-h recovery in LN229 parental and FBXO42 K/O cells. DNA damage markers detected: pCHK1 S296 (cell cycle checkpoint), pKAP1 S824, pRPA32 S33 (single-strand breaks), and γH2Ax (DNA DSBs). (G) Flow cytometry analysis of Edu and DAPI labelled LN229 and FBXO42 K/O cells (mean +/− SEM, n = 3). (H) Representative plot of flow cytometry analysis of Edu and DAPI labelled LN229 and FBXO42 K/O cells. Source data are available online for this figure.

PP4R3A, PP4R3B, or PP4R4, thus leading to the formation of four distinct oligomeric complexes (PP4C/PP4R1; PP4C/PP4R2/PP4R3A; PP4C/PP4R2/PP4R3B; PP4C/PP4R4) (Chen et al, 2008; Hwang et al, 2016; Lyons et al, 2021b). Elegant structural studies have defined the mode of substrate recognition operated by PP4R2 in complex with PP4R3A, recognising proline-rich residues in targeted substrates (Lipinszki et al, 2015; Ueki et al, 2019). The oligomeric composition and diversity of PPPs are required for substrate selectivity, but little is known about the substrate engagement strategy and the mechanisms to regulate oligomeric assembly of PP4.

Here, we identify the E3 ligase adaptor FBXO42 as a major regulator of the phosphatase PP4. Our study highlights that SCF^FBXO42 induces ubiquitination of PP4, regulating PP4R2 subunit assembly with PP4C, thus acting as a major inhibitor of PP4 phosphatase activity. Depletion of FBXO42 results in widespread effects on phosphorylation brought about by spurious activation of PP4. This mechanism highlights a yet undiscovered signalling node with multiple effects on cell survival and proliferation.

## Results

### Loss of FBXO42 induces multiple cellular defects including defective DNA damage response (DDR)

The advent of Clustered Regularly Interspaced Short Palindromic Repeats (CRISPR) has broadly enhanced our capacity for querying drug-gene interactions at a genome-wide scale. Several screens have been performed to identify genes contributing to cell viability upon the inhibition of the Ataxia telangiectasia and Rad3-related protein kinase (ATR) (Saldivar et al, 2017; Saldivar et al, 2018), a key controller of the DNA damage response (DDR) and regulator of mitotic entry at the G2/M checkpoint. A consensus set of vulnerabilities became evident from four genome-wide CRISPR screens using two ATR inhibitors (AZD6738 and VE-821) in multiple cell lines (Hustedt et al, 2019). FBXO42 was identified among the significant genes whose depletion increased cell death upon ATR inhibition using two inhibitors in the four indicated cell lines (Figs. 1A and EV1A) (Hustedt et al, 2019). A screen focused on the ubiquitin system, testing the genes mediating sensitivity to 41 compounds, identified FBXO42 as a vulnerability in cells treated with microtubule depolymerizing agents (colchicine) and BI-2536,

an inhibitor of the mitotic kinase Polo-like kinase 1 and other related kinases (Figs. 1B and EV1B) (Hundley et al, 2021). Furthermore, the loss of FBXO42 was shown to promote resistance to MEK inhibitors in melanoma (Nagler et al, 2020) and specifically impair the growth of a subset of glioblastoma stem-like cells (Hoellerbauer et al, 2024; Toledo et al, 2015). We had previously conducted a high-resolution CRISPR screen to identify genes controlling cell viability in response to treatment with ionising radiation (IR) (Yang et al, 2024), where FBXO42 was also among the hits noted (Fig. 1C). We clustered the genes scoring in the ATR inhibitors sensitivity, mitotic inhibitor sensitivity, and sensitivity to IR and compared the common hits. Only the FBXO42 gene scored in all the datasets (Fig. 1D), suggesting a pleiotropic function.

To exclude the possibility that FBXO42 is a frequent hit due to an off-target characteristic of the sgRNAs used in the screens, we generated two glioblastoma cell lines (LN229 and H4) where FBXO42 was knocked out using sgRNA sequences distinct from the library (Appendix Fig. S1). Both LN229 and H4 cell lines lacking FBXO42 were significantly more sensitive to VE-821 (ATRi) and indibulin (a microtubule depolymerising agent, analogue of colchicine), confirming the functional role of FBXO42 after ATR inhibition and mitotic spindle checkpoint activation identified in the genome-wide screens (Fig. EV1C–F). Cells lacking FBXO42 showed increased sensitivity to IR, while, upon expression of FBXO42 from an exogenous promoter, the sensitivity was restored back to control levels (Fig. 1E). The latter experiment shows that the sensitivity to IR is mediated specifically by the loss of FBXO42, excluding potential off-target or clonal effects of the FBXO42 knockout cells. The phenotype was also reproduced in a different glioblastoma cell line (H4) knockout for FBXO42 (Fig. EV1G). Upon analysing the phosphorylation events induced by IR treatment in cells lacking FBXO42, we observed a reduction in phosphorylation events controlling the DNA damage checkpoint (levels of pChk1 S345) after IR treatment. In addition, we noted a decrease in RPA32 phosphorylation at serines 4/8 [marker of ssDNA at DSB repair sites phosphorylated by Ataxia telangiectasia mutated (ATM) and DNA-dependent protein kinase (DNA-PK)] and serine 33 (marker of ssDNA phosphorylated by ATR) (Figs. 1F and EV1H). γH2Ax levels induced by IR were mildly impacted by depletion of FBXO42 (Fig. EV1G,F) but were induced upon increased expression of FBXO42 together with heightened phosphorylation of Chk1 (Fig. 1F). Cell lines lacking FBXO42 show alterations in cell cycle profiles, with a significant increase of cells in

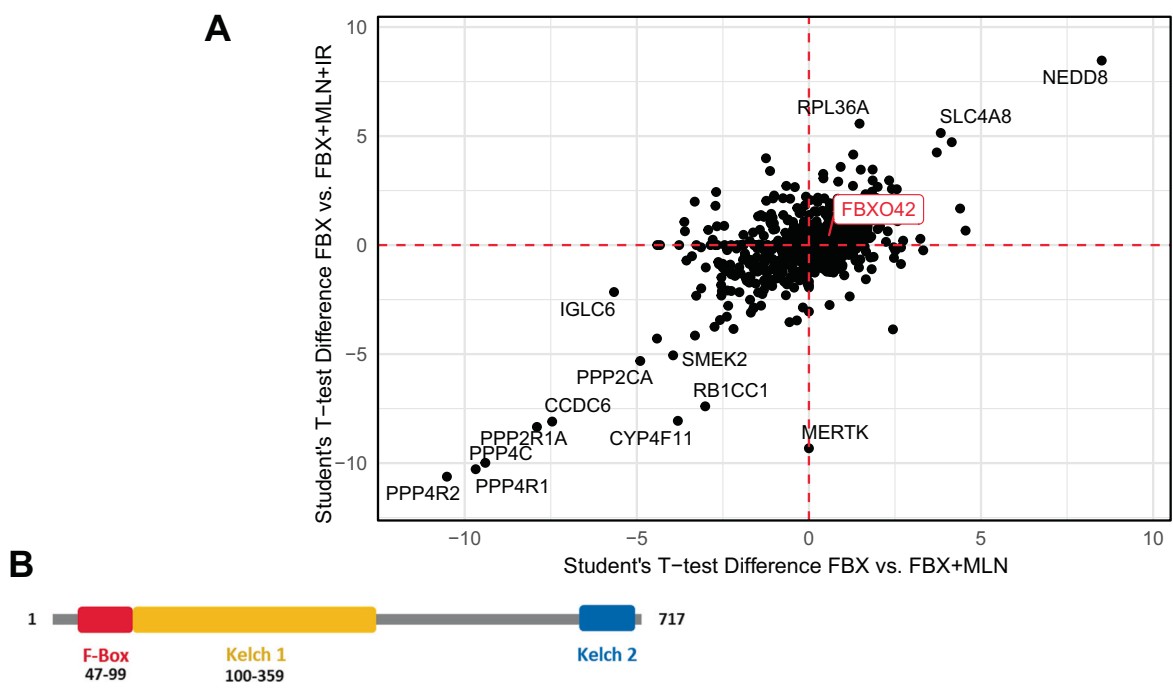

**A**

**B**

**C**

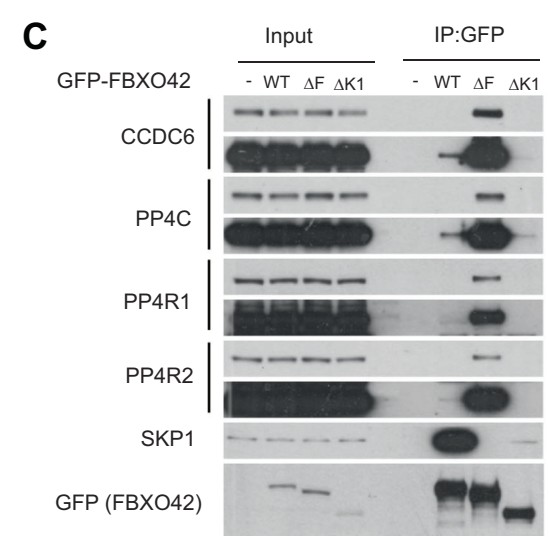

**D**

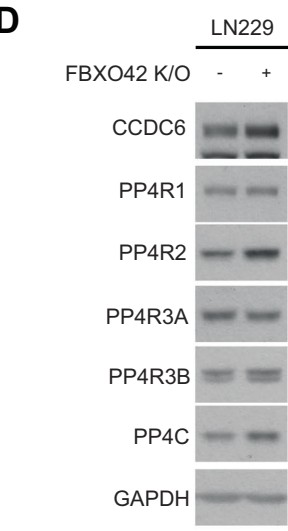

**E**

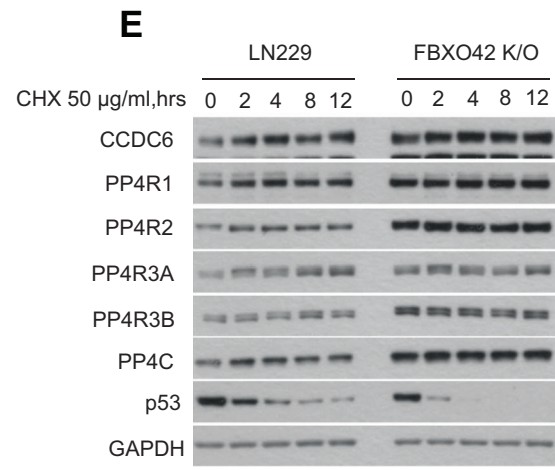

◄ **Figure 2. FBXO42 interacts with PP4 phosphatase.**

(A) Scatter plot of the log fold changes of FBXO42 IP-MS experiments +/− MLN4924 vs +/− ionisation radiation. The condition and relative level of the proteins is indicated in each quadrant. (B) Schematic representation of the domains in FBXO42. The F-box domain and Kelch domain are highlighted. The kelch domain is divided in two sections (Kelch1 and Kelch2) as reported on the PFAM database(http://pfam.xfam.org). (C) Immunoblotting after immunoprecipitation of GFP FBXO42 WT, FBXO42 F-box (ΔF) mutant and FBXO42 Kelch (ΔK1) mutant. (D) Immunoblotting of the indicated proteins in LN229 parental and FBXO42 K/O cells. (E) Immunoblotting of FBXO42 interactors after treating LN229 parental or FBXO42 K/O cells with cycloheximide (CHX) 50 μg/ml for the indicated times (hrs = hours). Source data are available online for this figure.

G1 and reduced cells entering S-phase, a defect which could be ascribed in part to altered S-phase checkpoint control and accumulation of cells with DNA damage in G1 (Fig. 1G,H). Thus, the overall reduction of phosphorylation events is indicative of a defect in the DNA damage response, which eventually results in a compromised DNA repair.

## FBXO42 interacts with PP4 phosphatase

The multitude of phenotypic effects induced by FBXO42 depletion could be explained by the fact that E3 ligases often target multiple substrates by recognizing degrons in proteins irrespective of the biological process they are involved in. A well-established example is the beta-transducin repeat containing E3 ubiquitin protein ligase (β-TrCP) which recognizes the DSGXXS degron once both serines have been phosphorylated (Guardavaccaro et al, 2003; Jin et al, 2003; Margottin-Goguet et al, 2003). The control of degradation is dictated by the kinases which phosphorylate the DSGXXS, with β-TrCP controlling multiple substrates involved in diverse cellular processes (Bi et al, 2021). We reasoned that FBXO42 might follow a similar principle and attempted to identify interactors of FBXO42 which are specifically impacted by IR, given its functional role in controlling the DDR. To isolate FBXO42 substrates, we treated cells with MLN4924 (a selective inhibitor of NAE1) (Soucy et al, 2009) and identified interactors after immunoprecipitation and liquid chromatography/mass spectrometry (LC/MS) as previously done for other CRLs (Burdova et al, 2019; Chen et al, 2022; Fung et al, 2018; Raducu et al, 2016). Neddylation is required for ubiquitination mediated by CRLs (Duda et al, 2008); thus, inhibition of NAE1 results in inactivation of CRL ubiquitination activity. Differential interacting partners enriched in the presence of MLN4924 are likely substrates, as their interaction with the adaptors is increased when the CRL-mediated ubiquitination is prevented. Surprisingly, IR treatment alone does not lead to major differences in interactions with FBXO42 (Fig. 2A, *lower right quadrant*). As expected, MLN4924 reduces the interaction of SCF^FBXO42 with Nedd8, judging by its enrichment in the top right quadrant (Fig. 2A). Most importantly, in the presence of MLN4924, FBXO42 binds more strongly to the protein phosphatase 4 (PP4) complex, including subunits PP4C, PP4R1, PP4R2, PP4R3A (SMEK1) and PP4R3B (SMEK2) together with CCDC6, a cofactor and substrate of PP4 implicated in the DNA damage response (Cerrato et al, 2018) (Fig. 2A). The full dataset is presented in Dataset EV1.

In a parallel approach, we established a cell line expressing FBXO42 fused to TurboID (an engineered biotin ligase that conjugates biotin to proteins) (Branon et al, 2018) to identify proteins in proximity to FBXO42 (Fig. EV2A). This approach, which exploits a nonspecific biotin ligase, has been used extensively to identify interacting partners of cellular proteins (May and Roux,

2019) with improvements made to enhance efficiency (Larochelle et al, 2019). In addition, in this approach we retrieved the PP4 phosphatase complex as an MLN4924 dependent interacting partner of FBXO42 (Appendix Fig. S2A). The full dataset is presented in Dataset EV2.

Like other F-box proteins, FBXO42 has an F-box domain at the N-terminus and a substrate recognition domain (Kelch domain) at the C-terminus (Fig. 2B) (Cardozo and Pagano, 2004). To establish the domains involved in the recruitment of the PP4 complex, we performed co-immunoprecipitation of either WT FBXO42 or mutants lacking either the F-box (ΔF) or the Kelch 1 (ΔK1) domain. As expected, ΔF mutants, although unable to bind Skp1 and Cul1, strongly bound to the PP4 phosphatase, thus suggesting that PP4 is a bona fide substrate of FBXO42 (Fig. 2C). The ΔK1 truncation abolished binding to PP4, confirming the need for the kelch domain in substrate engagement (Fig. 2C).

We initially hypothesized that FBXO42 targeted the PP4 phosphatase for ubiquitin-dependent degradation and measured the levels of PP4 phosphatase subunits in cells where FBXO42 had been knocked out by CRISPR. We noted a consistent increase in PP4C and PP4R2 levels upon FBXO42 depletion, while other subunits were not impacted (Fig. 2D). Simultaneously and surprisingly, no change in half-lives was noted for PP4 subunits in FBXO42 K/O cells (Fig. 2E). In the same settings, we also observed a reduction in the half-life of p53, which is mutated in LN229 cells. This observation has previously been reported, confirming the conditions of the experiment and effects due to FBXO42 loss (Hoellerbauer et al, 2024; Lu et al, 2024). The long half-lives of PP4 subunits we observe could be due to a cell-line-specific effect; thus, we measured PP4 half-lives in H4 cells harbouring a WT p53. Also, in this case, no change in half-lives was observed for PP4 subunits (Fig. EV2B). Finally, overexpression of FBXO42 (in HEK293T cells) did not lead to significant changes in the levels of PP4 phosphatase (Fig. EV2C).

## FBXO42 ubiquitinates PP4 phosphatase

Based on its enrichment upon MLN4924 treatment, along with unchanged half-lives in the absence of FBXO42, we predict PP4 complex to be ubiquitinated by FBXO42, although not targeted for degradation. To further explore the latter, we tested the ability of FBXO42 to ubiquitinate PP4 directly. Expression of FBXO42 induced a significant increase in ubiquitination of all components of the PP4R2 complex, including PP4C, PP4R2, PP4R3A and PP4R3B (Figs. 3A and EV3A). To confirm that the detected ubiquitin ladder is dependent on FBXO42 activity, we tested ubiquitination of PP4 using FBXO42 mutants mentioned above (ΔF and ΔK1; Fig. 2C). Both mutants were unable to ubiquitinate PP4 as the FBXO42 WT did (Fig. EV3B).

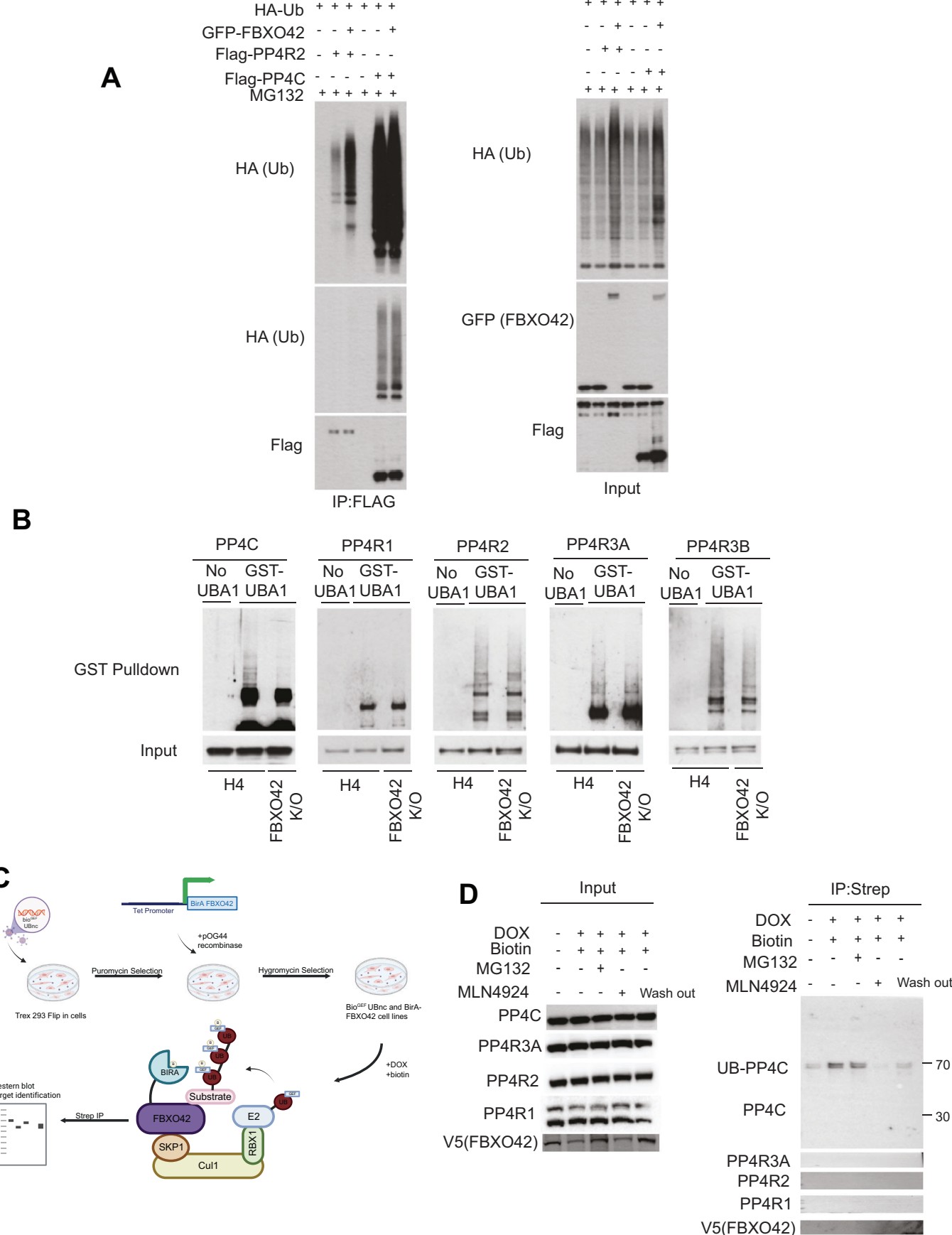

**Figure 3.  FBXO42 ubiquitinates the PP4 phosphatase.**

(A) Immunoblotting after expression of GFP-FBXO42, HA-ubiquitin and either Flag-PP4R2 or Flag-PP4C in HEK293T as indicated. Flag-PP4R2 or Flag-PP4C was isolated *via* Flag agarose beads pulldown under denatured conditions before immunoblotting. Input samples are in the right panel. (B) Immunoblotting after isolation of endogenous ubiquitinated proteins using recombinant GST-tagged UBA domain of UBQLN1 protein from H4 parental cells or cells FBXO42 K/O. Input samples before immunoprecipitation are indicated—bottom panel. (C) Schematic representation of the E-Stub proximity ubiquitin labelling and enrichment. BirA-FBXO42 was co-expressed with a biotin acceptor ubiquitin mutant (bio^GEF^UBnc) followed by biotin treatment. Nascent ubiquitinated substrates are thus biotinylated and immunoprecipitated. (D) Immunoblotting of PP4 complex after expression of (doxycycline-inducible) BirA–FBXO42 and biotin bio^GEF^UBnc-Ub in T-REx™ Hek293 cells. Proteins were induced by doxycycline for 24 h and supplemented with biotin for 3 h with or without MG132/MLN4924 as indicated, before protein lysis and immunoprecipitation. The "wash out" lane corresponds to removal of MLN4924 for 1 hour before collection of samples. Input samples indicated in the left panel. Source data are available online for this figure.

The non-physiological expression of E3 ligases and ubiquitin itself can lead to artefactual ubiquitination events. Particularly in the case of protein complexes, overexpression might force ubiquitination of components of the complex that are not modified physiologically. We therefore used two parallel approaches to establish the direct ubiquitination targets: we first sought to detect PP4 ubiquitination at endogenous levels in WT and FBXO42 K/O cell lines and followed this up with a recently developed proximity ubiquitination assay called E3-substrate tagging by ubiquitin biotinylation (E-Stub) (Huang et al, 2024). Upon unbiasedly enriching all ubiquitinated proteins from parental LN229 cells using the recombinant ubiquitin-associated domain (UBA domain) of the UBQLN1 protein in a ubiquitin-binding entity (UBE) pulldown assay (Fiil et al, 2013), we detected endogenous polyubiquitinated PP4 in H4 control cells and compared it to FBXO42 K/O (Fig. 3B). The same experiment was conducted in LN229 cell lines (Fig. EV3C). Taking together the results from LN229 and H4 on the different PP4 subunits in parental versus FBXO42 K/O lines, we could conclude that: PP4R1 was unlikely to be ubiquitinated; the PP4 complex most impacted by FBXO42 depletion was the one containing PP4C-PP4R2-PP4R3A; and finally, the most consistent effect across the two cell lines was the reduction of ubiquitylation observed on the PP4C subunit upon FBXO42 depletion (Figs. 3B and EV3C). Thus, we hypothesize that FBXO42 specifically ubiquitinates the PP4C subunit within the PP4R2 complex.

More evidence in support of the latter was revealed by our modified E-stub. In our experiment, we used a ubiquitin moiety with a biotin acceptor tag as described in (Barroso-Gomila et al, 2023; Merino-Cacho et al, 2025) and the E3 (FBXO42) with the biotin donor (BirA) (May and Roux, 2019). Both the acceptor and donor biotin modules are under the control of tetracycline in a Flp-in system line (Fig. 3C). Use of E-stub confirmed that we could retrieve ubiquitinated PP4C in proximity to the FBXO42 complex, but not PP4R1, PP4R2, and PP4R3A (Fig. 3D). The proximal ubiquitination of PP4C was dependent on the activity of FBXO42, as MLN4924 abolished the retrieval of PP4C. In addition, MG132 did not increase PP4C retrieval, further supporting that ubiquitination of PP4C did not lead to degradation. Interestingly, the detected PP4C band that was enriched by biotin, had a higher molecular weight than what was visible in the input, suggesting an attachment of perhaps three ubiquitin moieties, thus indicating a specific signalling event rather than a degradative signal (Fig. 3D). We tested the capacity of over-expressed FBXO42 to generate ubiquitin chains on PP4C and PP4R2 by using ubiquitin variants where only one lysine was available for conjugation, because the remaining lysines were

converted to arginine. We noted a preference for extending K33 type of chains on both PP4C and PP4R2 (Fig. EV3D,E). K33 chains have been assigned non-degradative functions, in support of a non-degradative ubiquitination mediated on PP4C (Agrata and Komander, 2025). The caveat of this experiment is over-expression of both E3 ligase and substrates, which might result in spurious ubiquitination events for clearing excess proteins. In summary, the main interactor and substrate of FBXO42 is the PP4 phosphatase, suggesting the possibility that multiple phenotypic effects attributed to FBXO42 might be due to its modulation of this versatile phosphatase.

## FBXO42 is a major regulator of PP4 phosphatase

We have thus far observed that FBXO42 ubiquitinates PP4C and is likely to regulate the activity of PP4 complex rather than the levels. For a comprehensive understanding of the effect of FBXO42 on phosphorylation, we performed a mass spectrometry based phosphoproteomic analysis in LN229 cells (parental and FBXO42 K/O). We also conducted a phosphoproteomic analysis of cells in which PP4C was reduced by siRNA. Given the complexity of these datasets, we will describe these in sequence by first looking at total protein levels, then phosphoproteins, and finally comparing findings in FBXO42 knockout cells to cells with reduced PP4C levels.

About ~300 proteins were either upregulated or downregulated (assuming a *p*-value of 0.05 and a log2 fold change of at least 1-fold) in FBXO42 knockout cells (Fig. EV4A), likely an indirect consequence of a broader dysregulation of phosphorylation. The levels of PP4 phosphatase components were not majorly impacted in this dataset (Fig. EV4A). A gene enrichment analysis was conducted to highlight the pathways being downregulated in FBXO42 K/O, using EnrichR and selecting the MSigDB hallmarks 2020 enrichment tool (Fig. EV4B). This analysis highlighted significant alterations of genes involved in G2/M checkpoint, mitotic spindle and DNA repair in line with the phenotypes described above and previous publications (Hoellerbauer et al, 2024; Hundley et al, 2021; Nagler et al, 2020; Toledo et al, 2015). The pathways enriched in cells knockout for FBXO42 include oxidative phosphorylation and epithelial-mesenchymal transition (Fig. EV4C), important cancer-associated phenomena, which require further investigation, using appropriate cancer models. Total proteomic analyses of proteins changed in cells depleted of PP4C by siRNA did not show protein alterations (Fig. EV4D). The finding could be due to the experimental conditions where we could only reduce PP4C levels before starting cellular attrition (Fig. EV4F).

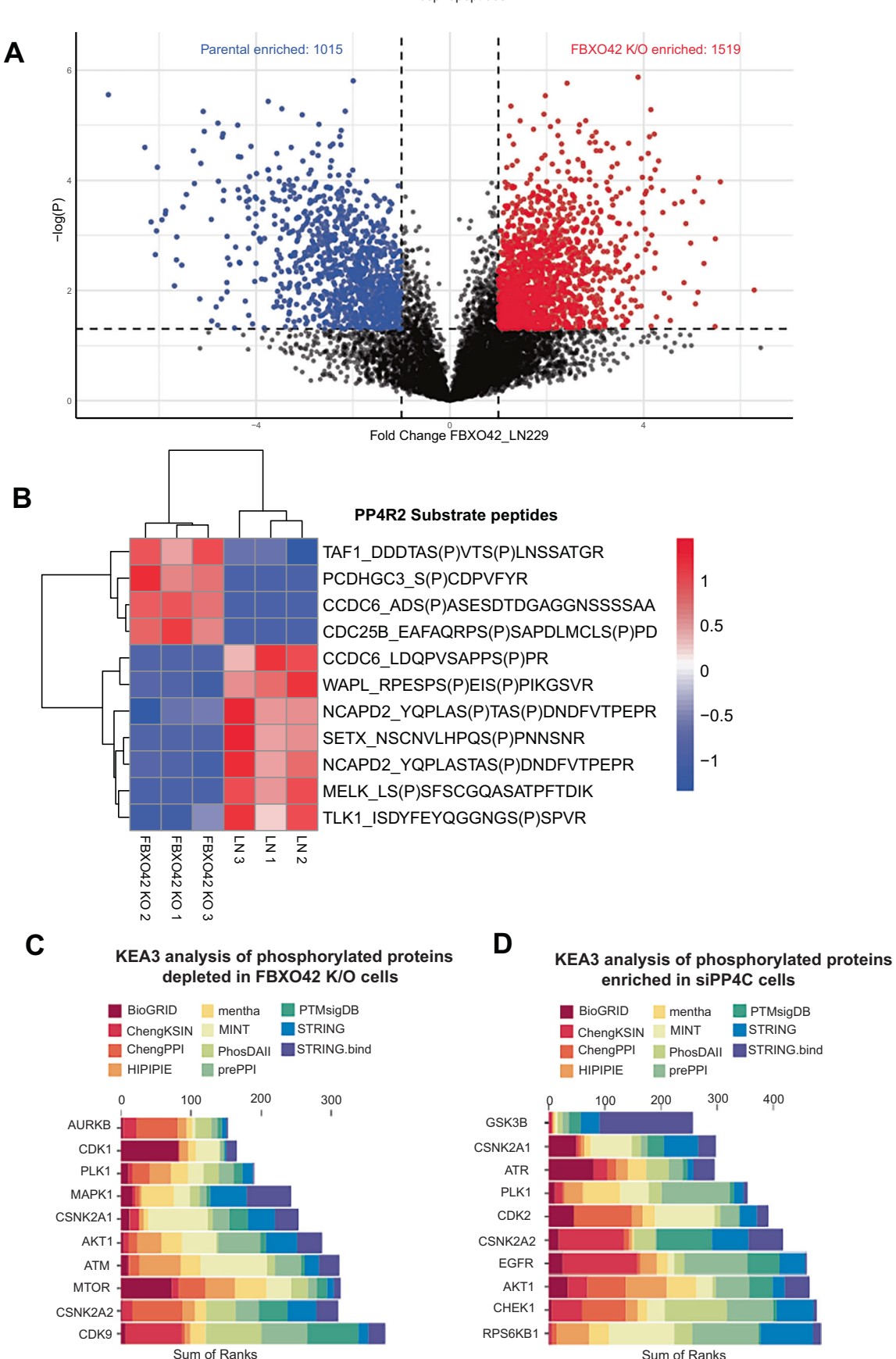

**Figure 4.    FBXO42 is a major regulator of PP4 phosphatase.**

(A) Volcano plot depicting differentially enriched phosphorylated motifs/phosphopeptides in LN229 parental cells compared to FBXO4 K/O cells. Peptides were considered significant if the *p*-value is greater or equal to 0.05 and fold change is greater or equal to 2. Significant proteins are highlighted in red or blue. (B) Heatmap of phosphopeptides in LN229 cells and FBXO42 K/O cells. The peptides from proteins considered to be substrates of the PP4R2 subcomplex previously annotated by (Ueki et al, 2019) and considered be significant *p*-value greater or equal to 0.05 and fold change greater or equal to 1. (C) Kinase enrichment analysis using KEA3 of proteins whose phosphorylation sites decreased in FBXO42 K/O cells when compared to LN229 parental. (D) Kinase enrichment analysis using KEA3 of proteins whose phosphorylation sites increased in cells treated with siRNA against PP4C compared to LN229 treated with siRNA control.

In the phospho-proteomic dataset we observed ~2500 phosphopeptides being differentially regulated, with 1015 peptides enriched in parental cells and 1519 enriched in FBXO42 knockout cells (downregulated in parental cells) (Fig. 4A). Studies on broad identification of PP4C substrates are lacking and only one study thus far has highlighted proline rich sites (FXXP and MXXP) as recognition substrates for the PP4C-PP4R2-PP4R3A/B (PP4R2/R3) complex (Ueki et al, 2019).

Thus, we matched the identified and validated 18 PP4R2/R3 substrates in Ueki et al (2019) with proteins whose phosphorylation changes upon FBXO42 K/O. Since our hypothesis is that FBXO42 is needed to restrain PP4C activity, FBXO42 cells should have high PP4 activity and reduced phosphorylation events. We observed that out of the 18 substrates of PP4R2/R3 reported, nine were not in the dataset, two were not differentially regulated and seven were downregulated in FBXO42 K/O (Fig. 4B). In other words, circa 60% of the known PP4R2 substrates, identified in our study, are dephosphorylated in FBXO42 K/O. CCDC6 presented two sites of which one was more phosphorylated in the FBXO42 knockout, and the other was less phosphorylated. Interestingly, the downregulated phosphorylation site on CCDC6 corresponds to a site which is in proximity to a canonical FXXP site. NCAPD2, whose phosphorylation is also reduced in the FBXO42 knockout, has an FXXP site retained within the tryptic peptides, bearing the phosphorylations.

The dataset where we reduced PP4C levels by siRNA had ~600 differential phosphorylation sites with only 4 PP4R2/R3 substrates being identified (Fig. EV4F,G), probably a reflection of the partial knockdown of PP4C and further contribution from PP4R1. Thus, the dataset cannot be used to derive bona fide PP4C substrates, but it could be used as a proxy of altered PP4C activity.

Gene enrichment analysis of proteins whose phosphorylation was reduced in FBXO42 knockout revealed mitotic progression and the G2/M checkpoint (Fig. EV4H). The curbing of PP4C activity by siRNA should reduce phosphatase activity and increase phosphorylation events. Gene enrichment analysis of the proteins being more phosphorylated in PP4C siRNA, reveals enrichment of mitotic progression and G2/M checkpoint (Fig. EV4I), equal to the enrichment of proteins whose phosphorylation is reduced in FBXO42 knockout (Fig. EV4H). Thus, although the phosphorylation sites are not the same between FBXO42 knockout and siRNA of PP4C, there is significant overlap of the pathways being affected. To provide a comprehensive view of the proteins being altered and common between the datasets, we report the sites in Appendix Fig. S2.

The main challenge in conducting this type of analyses is a dearth in our understanding of specific substrates of PP4. The activity of kinases is much better studied, given the availability of specific chemical inhibitors and several studies to define their consensus phosphorylation sites (Cohen et al, 2021; Johnson et al,

2023). In our model loss of FBXO42 leads to activation of PP4 phosphatase and, thus, decreased phosphorylation. We queried the kinases whose activity decreases in FBXO42 K/O cells by using Kinase Enrichment Analysis 3 (KEA3), an online tool to infer the activity of kinases (https://maayanlab.cloud/kea3/) (Kuleshov et al, 2021). The kinases whose activity was reduced, counteracted by PP4, were AURKB, CDK1, PLK1, MAPK1, CSNK2A1, AKT1, ATM, MTOR, CSNK2A2 and CDK9 (Fig. 4C). The reduction in kinase activity is in line with the phenotypes associated to FBXO42 depletion, and these kinases phosphorylate known PP4 substrates (Park and Lee, 2020).

To prove an additional association with the activity of PP4, we compared the kinase activity profiles from FBXO42 K/O cells to cells in which PP4 activity is curbed by siRNA. In this case we input a list of proteins whose phosphorylation was increased in PP4C siRNA into KEA3. The kinases thus identified were GSK3B, CSNK2A1, ATR, PLK1, CDK2, CSNK2A2, EGFR, AKT1, CHEK1 and RPS6KB1 (Fig. 4D). Out of the 10 kinases identified in FBXO42 K/O or PP4C siRNA, five were identical, and the others had very closely related profiles (Fig. 4C,D), again a strong indication that PP4 activity is altered in these cells. The lack of a stronger, or perhaps even a complete overlap, could be attributed to technical limitations, including only a partial depletion of PP4C, the presence of indirect effects, and/or the presence of additional substrates modulated by the other regulatory subunits PP4R1 and PP4R4, which are also impaired by PP4C depletion. The full dataset is presented in Dataset EV3.

## FBXO42 restrains PP4 activity to control survival after DNA damage

The phosphoproteomic dataset suggests an alteration of PP4 phosphatase activity in FBXO42 knockout cells; however, it is unclear whether a direct relationship can be established between FBXO42 depletion and increased PP4 activity. The reduced phosphorylation events we describe after DNA damage by IR in Figs. 1F and EV1H are PP4 dependent (Chowdhury et al, 2008; Lee et al, 2012; Lee et al, 2010; Nakada et al, 2008; Villoria et al, 2019; Zheng et al, 2019); thus, we aimed to understand whether phosphorylation events occurring upon IR treatment in FBXO42 lacking cells could be rescued by concomitant depletion of PP4C. FBXO42 K/O cells showed a significant decrease of RPA phosphorylation (Fig. 1F,G), which are known to be controlled by PP4 activity. The phosphorylation statuses of RPA and Chk1 is rescued by depletion of PP4C in FBXO42 K/O (Fig. EV5A). Interestingly, the depletion of PP4R1 did not affect these phosphorylation events as much as the depletion of PP4R2 (Fig. 5A). Most importantly, the increased IR sensitivity in cells knockout for FBXO42, can be fully rescued by depletion of PP4R2

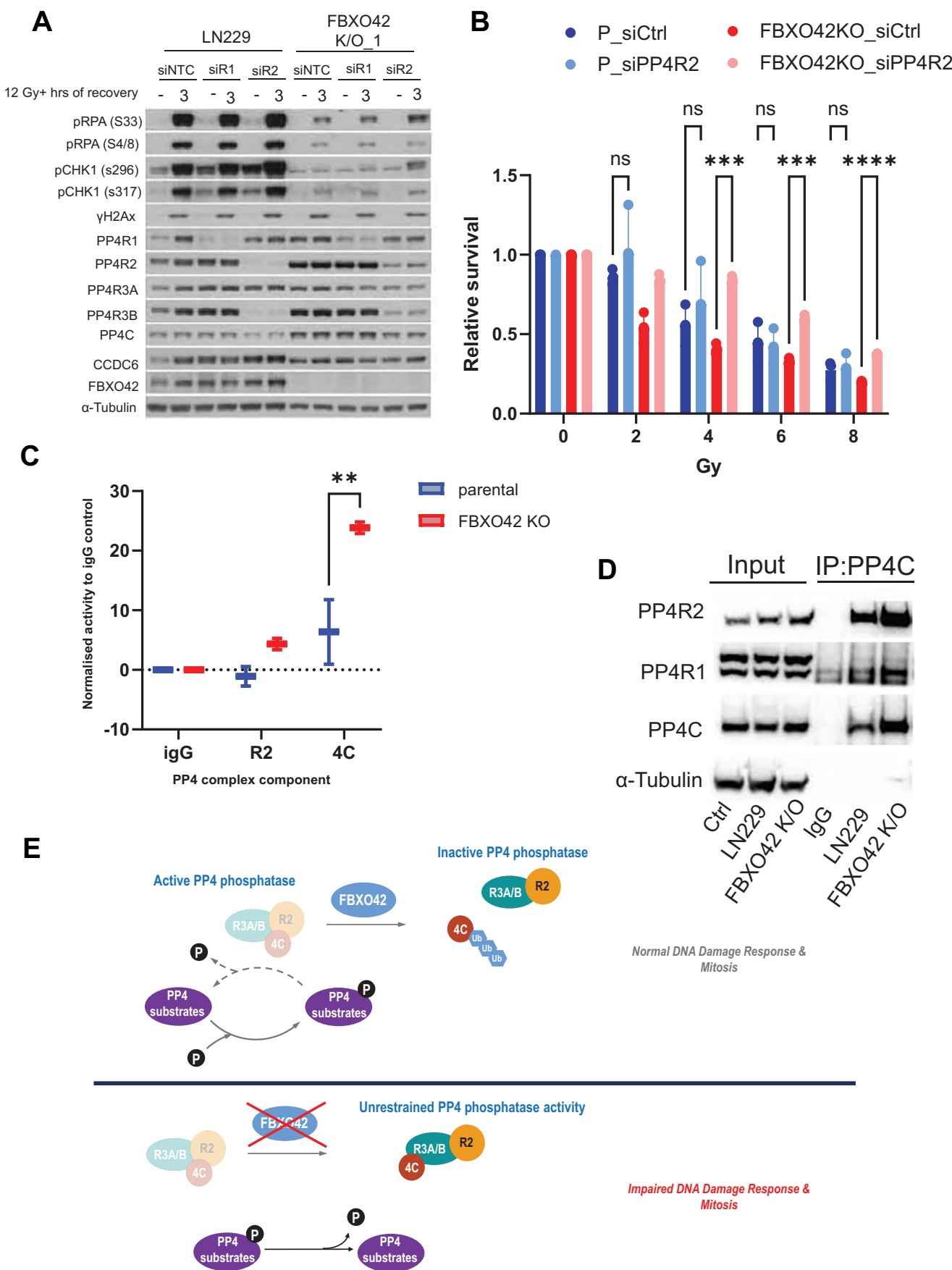

**Figure 5. FBXO42 restrains PP4 activity to control survival after DNA damage.**

(A) Immunoblotting of DNA damage markers and PP4 complex subunits in LN299 parental and FBXO42 K/O cell lines after IR treatment and 3-hour recovery with or without siRNA knockdown of PP4R1 and PP4R2. DNA damage markers detected: pRPA32 S33 (single-strand breaks), pRPA32 S4/8 (ssDNA at DSB), pCHK1 S296 (cell cycle checkpoint), pCHK1 S317 and γH2Ax (DNA DSBs). (B) Representation of relative survival after colony formation assay in LN229 cells and LN229 FBXO42 K/O cells treated with siRNA against PP4R2 compared to siRNA control. LN229 parental cells, LN229 FBXO42 K/O cells treated with siRNA against PP4R2, or siRNA control were seeded for colony formation assay and challenged with the indicated dose of IR. Seven days after IR, cells were stained with crystal violet and counted. Data presented as mean $+/-$ SD; $n = 3$ biological replicates. Two-way Anova was performed as statistical analysis. Exact $P$ values left to right are 0.83, 0.86, 0.0004, 0.9, 0.0003, 0.9, <0.0001. *$P \le 0.05$; **$P \le 0.01$; ***$P \le 0.001$; ****$P \le 0.0001$. (C) Box and whisker plot of phosphatase activity of isolated PP4 complex isolated from LN299 and FBXO42 K/O cells. The activity of complexes isolated by endogenous IP of PP4R2 and PP4C are shown and represents three biological replicates. Two-way Anova was performed as statistical analysis. Exact P value 0.009. *$P \le 0.05$; **$P \le 0.01$; ***$P \le 0.001$; ****$P \le 0.0001$. (D) Immunoblotting of the indicated proteins after immunoprecipitation of PP4C from LN229 and FBXO42 K/O cells. Input samples are indicated on the left. (E) Schematic depiction of the regulatory activity of FBXO42 on the PP4 complex. Source data are available online for this figure.

(Fig. 5B). This experiment further consolidates the model whereby aberrant PP4 activity is brought about by the loss of FBXO42, establishing a causal relationship between loss of FBXO42, increased PP4 activity and defective DNA damage response/DNA repair.

To further ascertain increased activity of the PP4 phosphatase, we measured the activity of the phosphatases on a model PP4 substrate (KRpSIRR), comparing parental cells to cells knockout for FBXO42. FBXO42 cells showed a significant increase of PP4C and PP4R2 activity (Fig. 5C), although levels of PP4C and PP4R2 were normalised across experimental samples (Fig. EV5C).

An increased activity of PP4 phosphatase can be brought about by altered localisation of the PP4C/PP4R2 complexes and/or altered assemblies of the regulatory and catalytic subunits. While we did not observe differences in PP4C/PP4R2 localisation (Appendix Fig. S3), we observed an increase in the interaction of PP4C with PP4R2 upon depletion of FBXO42 in two experiments presented below. We immunoprecipitated PP4C from parental and FBXO42 K/O cells and measured the relative abundance of the PP4 associated subunit using LC/MS. We observed that the lack of FBXO42 increased the interaction of PP4R2 with PP4C by more than two-fold (Fig. EV4E), but did not have the same impact on the interaction of PP4C with PP4R1, which remains the same or is possibly reduced. The full dataset is presented in Dataset EV4. To consolidate multiple observations obtained so far, we also immunoprecipitated endogenous PP4C in parental and FBXO42 K/O cells, confirming an increase in the interaction between PP4C and PP4R2 (Fig. 5D). Similar results were obtained after immunoprecipitation of PP4R3A and PP4R3B (Fig. EV5E)—these sub-complexes being the true effectors of the observed phenotypes in FBXO42 knockout.

## Discussion

Here we identified an uncharacterised regulator of PP4, providing insights into an unprecedented regulatory mechanism. Our work indicates that the main function of FBXO42 is to restrict the activity of the PP4 phosphatase, particularly the PP4R2 complex. The mechanism is attributed to a likely non-degradative ubiquitination event on PP4C triggered by FBXO42 to restrict the formation of active PP4C/PP4R2 complexes. Lack of FBXO42 unleashes uncontrolled PP4 activity, thus incapacitating the action of kinases involved in cell cycle control and signalling, as shown by our phosphoproteomic approach. We observe that increased

activity of PP4 in FBXO42 K/O results in defective DNA damage checkpoint signalling and reduced CHK1 and RPA phosphorylation. These events can be rescued by the depletion of PP4 in an FBXO42 K/O background, establishing a direct relationship between FBXO42 and PP4 in modulating the DNA damage response (Fig. 5E). A manuscript with parallel observations points out that the survival of cancer cells depleted of FBXO42 can be reverted by concomitant depletion of PP4, solidifying the notion that FBXO42 acts as a major regulator of PP4 (preprint: Spangenberg et al, 2025). The type of ubiquitination assembled by FBXO42 (on PP4) is likely to be non-degradative, but it is not possible to exclude that smaller pools of PP4C and/or PP4R2 are degraded in cells, thereby altering the stoichiometry of the PP4 subunits with the net effect of increasing PP4 activity. In the case of PP2A, at least five regulatory proteins are in place to assist the biogenesis of the full complex and restrict the activity of the catalytic subunit (alpha4, PTPA, LCMT1, PME-1, and TIPRL) (Fellner et al, 2003; Guo et al, 2014; Sents et al, 2013; Stanevich et al, 2011; Wu et al, 2017; Xing et al, 2008). FBXO42 might assist in the biogenesis of PP4 complexes to restrict activity until the full complex is assembled. Indeed, while classically C-terminal methylation of the catalytic subunit is required to assemble the PPP complexes, this is less important in the assembly of PP4C/PP4R2, where FBXO42 is playing a major role (Hwang et al, 2016; Lyons et al, 2021a). It is also worth to point out that FBXO42, PP4R2, PP4R3A and PP4R3B are localised in the nuclei, while the PP4R1 subunit is cytoplasmic, suggesting that FBXO42 is mostly acting on the complexes formed through PP4R2 (Appendix Fig. S3) (Bekker-Jensen et al, 2017).

Generally, we cannot exclude that other PPP complexes are controlled by FBXO42, as we retrieve other phosphatases (PP2 and PP6) in our interaction proteomic studies (Figs. 2A and EV2A).

We provide initial evidence in FBXO42 K/O cells that the assembly of PP4C to the regulatory subunits is defective. However, further studies are required to establish the type of polyubiquitin chains formed by FBXO42 and how these chains mechanistically prevent the formation of PP4C/PP4R2 complexes. Interestingly, PP4C K300 is found to be ubiquitinated, and it resides at a critical contact point of interaction between the catalytic and regulatory subunits. It is tantalising to speculate that FBXO42 might catalyse ubiquitination of this residue to prevent assembly of the catalytic subunit with the regulatory subunit. However, a full set of experiments is complicated by the essential role of PP4 in cell survival and the requirement for the K299/300 residues in PP4C to form hydrogen bonds with the PP4R2 and PP4R3A/B subunits.

A recent study suggests RBPJ to bea substrate for FBXO42, regulated through non-degradative K63 ubiquitination (Jiang et al, 2022). We did not identify RBPJ in our interaction studies. However, the finding supports the idea that FBXO42 catalyses non-degradative ubiquitin chains. It is unclear how mechanistically F-boxes can trigger non-canonical ubiquitination having a common E2 recruiter in Rbx1. It may be conceivable that super complex assemblies with E3s trigger an initial ubiquitination event, and other E3s catalysing chain extension might be in play (Scott et al, 2016). The paucity of reagents to detect non-canonical ubiquitination strongly limits the capacity of further elucidating the details of the regulatory events controlled by non-degradative polyubiquitin chains.

Our work outlines a central regulatory node to control survival, with several cancer-associated phenotypes requiring further studies. FBXO42 inhibition leads to dysregulated kinase activity and might exacerbate a latent liability of cancer cells, promoting cell death.

# Methods

### Reagent and tools table

| Reagent and Resource | Source | Identifier |
| --- | --- | --- |
| **Antibodies** | | |
| Anti-Flag agarose beads (affinity agarose gel) | Sigma-Aldrich | A2220-5ML |
| GFP-trap agarose beads | Proteintech | gta-20 |
| Mouse anti-FBXO42 | ORIGENE | TA800210 |
| Mouse anti-CUL1 | Invitrogen | 32-2400 |
| Mouse anti-α-tubulin | Santa Cruz | sc-23948 |
| Mouse anti-GAPDH | Invitrogen | MA5-15738 |
| Mouse anti-GFP | Santa Cruz | SC-9996 |
| Mouse anti-HA | Biolegend | 901501 |
| Mouse anti-V5 | eBioscience | 14-6796-82 |
| Rabbit anti-PPP4R1 | Bethyl | A300-836A |
| Rabbit anti-PPP4R2 | Bethyl | A300-838A |
| Rabbit anti-PPP4R3A | Bethyl | A300-840A |
| Rabbit anti-PPP4R3B | Bethyl | A300-842A |
| Rabbit anti-PPP4C | Bethyl | A300-835A |
| Anti-PPP4C antibody | St John's Laboratory | STJ25096-100 |
| Rabbit anti-CCDC6 | Atlas | HPA019051 |
| Rabbit anti-Flag | Sigma-Aldrich | F7425-.2MG |
| Rabbit anti-SKP1 | Cell Signaling Technology | 2156S |
| Rabbit anti-γH2Ax | Novus Biologicals | NB100-2280 |
| **Reagents** | | |
| 0.5 M EDTA pH 8.0 solution | AppliChem | A48920500 |
| 1 M Magnesium chloride stock | Sigma-Aldrich | 63069-100 ML |
| 1 M Tris-HCl pH 7.5 stock | VWR | E691-500ML |
| 1,10-Phenanthroline | Scientific Laboratory Supplies | CHE2730 |

| Reagent and Resource | Source | Identifier |
| --- | --- | --- |
| 4-12% Bis-Tris gradient precast gel | Thermo Scientific | NW04125BOX |
| 4X Laemmli sample buffer | Invitrogen | NP0008 |
| 5 M NaCl stock | Gibco | 24740011 |
| Acetic Acid Glacial | Sigma-Aldrich | A6283-500ML |
| Ammonium bicarbonate | Fluka Analytical | 09830-500G |
| Antifade mounting medium | VECTASHIELD | H-1000 |
| Beta-glycerophosphate | Sigma-Aldrich | G5422 |
| Beta-mercaptoethanol | Thermo Scientific | 31350010 |
| Blasticidin S HCl | Cambridge Bioscience | 14499-25 mg-CAY |
| Biomol® Green | Enzo Life Sciences | BML-AK111-0250 |
| Crystal violet | Alfa Aesar | B21932 |
| Click-iT™ EdU Alexa Fluor™ 647 Imaging Kit | Thermo Scientific | C10340 |
| Cycloheximide | Sigma-Aldrich | C7698-5G |
| DAPI | Sigma-Aldrich | MBD0015-1ML |
| Doxycycline | Sigma-Aldrich | D9891 |
| Dulbecco's modified Eagle medium | Sigma-Aldrich | D6429-500ML |
| Ethanol | Sigma-Aldrich | 32221-2.5L-M |
| Foetal bovine serum | Gibco | 10500064 |
| Glutathione sepharose beads | Cytiva | GE17-0756-01 |
| Glycerol | Thermo Scientific | 17904 |
| HEPES | Sigma-Aldrich | H3375-500G |
| Hygromycin B | Thermo Scientific | 10687010 |
| KCl | Sigma-Aldrich | P9541-1KG |
| Lambda phosphatase | New England Biolabs | P0753S |
| Lipofectamine 2000 | Invitrogen | 11668019 |
| Methanol | Fisher Scientific | 10164663 |
| N-Ethylmaleimide (NEM) | Sigma-Aldrich | E3876-5G |
| NaF | Fluka Analytical | S7920-100G |
| Nitrocellulose membrane | Amersham | 10600006 |
| Nocodazole | Selleckchem | S2775 |
| Nonidet P-40 alternative | EMD Millipore | 492016-100 ML |
| Okadaic acid | Cayman Chemical | 10011490-50 ug-CAY |
| Penicillin-streptomycin | Gibco | 15140122 |
| Phosphoric acid | Sigma-Aldrich | 695017-100 ML |
| PIPES | Sigma-Aldrich | P1851 |
| PMSF | Santa Cruz | sc-482875 |
| Polyethylenimine linear (PEI) | Polysciences Inc. | 23966 |
| PR-619 | ApexBio | A821 |
| Protease inhibitor cocktail for mammalian | Sigma-Aldrich | P8340 |
| Protein A/G agarose beads | Santa Cruz | sc-2003 |
| Puromycin | Santa Cruz | sc-108071 |

| Reagent and Resource | Source | Identifier |
|---|---|---|
| Sodium dodecyl sulfate (SDS) | Sigma-Aldrich | 74255-250G |
| Strep-Tactin™ Superflow™ High Capacity Resin | IBA Lifesciences | 2-1208-002 |
| Sucrose | Sigma-Aldrich | S0389-500G |
| Thymidine | Alfa Aesar | B21280 |
| Triethylammonium bicarbonate (TEAB) | Thermo Scientific | 90114 |
| Trifluoroacetic acid (TFA) | Fisher chemical | T/3258/PB05 |
| Tris(2-carboxyethyl)phosphine hydrochloride (TCEP) | Thermo Scientific | 77720 |
| Triton X-100 | Sigma-Aldrich | T8787-250ML |
| Tween 20 | Fischer Scientific | BP337-500ML |
| Cell lines | | |
| LN229 | ATCC | CRL-2611 |
| H4 | ATCC | HTB-148 |
| HEK293T | ATCC | CRL-3216 |
| Flp-In™ T-REx™ 293 Cell Line | Thermo Fisher | R78007 |

## Reagents, antibodies, and cell lines

All reagents, Antibodies and cell lines used in this study are listed in the relevant reagents and tools section.

## Cell culture

LN229, HEK293T and H4 were obtained from American Type Culture Collection (ATCC). All cell lines were cultured in Dulbecco's modified Eagle's medium (DMEM; Sigma-Aldrich, D6429-500ML) containing 10% foetal bovine serum (FBS; Life Technologies, 10500064) and the mixture of penicillin (100 U/ml) and streptomycin (100 µg/ml; Life Technologies, 15140122). LN229 and H4 cells knockout for FBXO42 were generated using CRISPR (sgRNA sequences: table above).

## Colony formation assay

Cells were counted and seeded at 400 cells per well in six-well plates and cultured for 6 h before being challenged with the indicated IR doses using a closed source gamma irradiator. After IR treatment, cells were allowed to propagate for 7 days (for HeLa-derived cell lines) or 14 days (for LN229-derived cell lines). Colonies were fixed and visualized using crystal violet solution (50% methanol, 10% ethanol, and 0.3% crystal violet) for 10 min at room temperature with gentle agitation (10 to 20 rpm). After rinsing with water and airdrying, colonies were counted using GelCount mammalian-cell colony counter (Oxford OPTRONIX). All colony formation assays presented in this study have been repeated at least three times and error bars represent SD of three biologically replicates.

## CRISPR knockout

Stable knockout was generated by co-transfecting Cas9 protein and three sgRNAs targeting the same gene, designed by Synthego, using Lipofectamine CRISPR Max (Life Technologies, CMAX00001) according to the protocol here: https://www.synthego.com/products/crispr-kits/synthetic-sgrna. Four days after transfection, cells were trypsinized and seeded as single cells in 96-well plates to isolate single clones. Cells in wells with proliferative clones were then trypsinized and expanded in bigger vessel until there were enough cells for immunoblotting. Clones that showed clear knockout were further validated by genomic DNA extraction and PCR amplification of the exon targeted by the sgRNAs. The PCR product was ligated into pCR4 vector using a TOPO TA cloning kit (Invitrogen, 450030). After transforming into DH5α competent cells, 10 colonies were picked and sent for Sanger sequencing.

## Generation of TurboID–FBXO42 stable cell line

Flp-In™ T-REx™ HEK293 cells (Invitrogen, R78007) containing a single genomic FRT site and stably expressing the Tet repressor were cultured in DMEM supplemented with 10% FBS, zeocin (100 µg/ml), and blasticidin (15 µg/ml). The medium was exchanged with fresh medium containing no antibiotics before transfection. For cell line generation, Flp-In HEK293 cells were co-transfected with the pCDNA3–TurboID–FBXO42 plasmid and the pOG44 Flp-recombinase expression vector (Invitrogen, V600520) for coexpression of the Flp-recombinase using Lipofectamine 2000 transfection reagent (Invitrogen, 11668019). Two days after the transfection, cells were selected in hygromycin-containing medium (100 µg/ml) for 2 to 3 weeks. To validate the TurboID–FBXO42 expression, cells were cultured in media containing doxycycline (1.3 µg/ml) for 24 h to induce TurboID–FBXO42 expression before immunoblotting.

## MS sample preparation for TurboID–FBXO42 pulldown

When TurboID–FBXO42 Flp-In™ T-REx™ HEK293 cells grown in 15-cm dishes reached 80% confluency, doxycycline (1.3 µg/ml) was added for 24 h to induce the expression of TurboID–FBXO42. Cells were further incubated with 50 µM biotin for 3 h to label proteins that came into close proximity with TurboID–FBXO42 in cells. Cells were harvested by scraping and washed three times with phosphate-buffered saline (PBS). For streptavidin pulldown of all biotin-labelled proteins (potential FBXO42 interactors), cell pellets were thoroughly resuspended in 1 ml of RIPA buffer [50 mM Tris-HCl (pH 8.0), 150 mM NaCl, 1% Triton X-100, 1 mM EDTA, and 0.1% SDS with protease inhibitor cocktail (Sigma-Aldrich, P8340)] and incubated on ice for 15 min. Insoluble material was removed by centrifugation. Cleared lysates were then incubated on a rotating wheel at 4 °C with 50-µl pre-equilibrated Strep-Tactin® Superflow® high-capacity resin (IBA, 2-1208-002) for 1 h. The suspension was then loaded on a Mini Bio-Spin Columns (Bio-Rad, 732-6207) to collect the beads. The beads were washed two times with 1 ml of RIPA buffer, three times with HNN buffer [50 mM HEPES (pH 7.5), 150 mM NaCl, and 50 mM NaF], and two times with 100 mM $NH_4HCO_3$ solution before being transferred to 2-ml Eppendorf tube in 400 µl of $NH_4HCO_3$ solution. For proteolysis, the sample was centrifuged at 200 g for 1 min to remove supernatant. Beads were resuspended in 100 µl of 8 M Urea in 100 mM $NH_4HCO_3$ solution and incubated at 20 °C for 20 min. Cysteine bonds were reduced with a final concentration of 5 mM tris(2-carboxyethyl) phosphine hydrochloride (TCEP) for

30 min at 37 °C and alkylated in a final concentration of 10 mM iodoacetamide for 30 min at room temperature in the dark. Beads were then proteolyzed with trypsin/Lys-C Mix (Promega, V5071) at a 25:1 protein:protease ratio (w/w) for 4 h at 37 °C on an orbital shaker. Urea concentration was then reduced to 1 M via adding 100 mM NH$_4$HCO$_3$ solution to the sample. Samples were digested overnight at 37 °C on an orbital shaker. Samples were desalted on C18 spin columns (Thermo Fisher Scientific, 89870) and washed according to the manual provided by the manufacturer and eluted with 0.1% trifluoroacetic acid (TFA) and 65% acetonitrile. Peptides were then dried in a SpeedVac vacuum concentrator and resuspended in 0.1% TFA and 2% acetonitrile in MS-grade water for MS analysis.

## Sample preparation for Flag-IP

FBOX42-flag interactors were eluted from the beads using 3x flag peptide in 50 mM Tris, 1 mM EDTA, 5 mM MCl$_2$ and 0.1% NP40. Eluted fractions were digested using S-Trap™ micro columns following manufacturer's protocol (ProTifi). Briefly, SDS was added to the samples (5% final concentration) before they were sequentially reduced with DTT (20 mM final concentration) and alkylated with Iodoacetamide (40 mM final concentration) for 30 min in the dark at room temperature. Samples were then acidified with phosphoric acid (1.2% final concentration) and proteins precipitated by adding 90% methanol in 100 mM TEAB buffer (1 to 7, sample: buffer ratio). Samples were loaded into the S-Trap™ micro columns cartridge and washed four times with 90% methanol in 100 mM TEAB buffer. Two µg of trypsin in 50 mM TEAB buffer was added into the S-Trap™ micro columns and incubated overnight at 37 °C. Finally, tryptic peptides were sequentially eluted from the S-Trap™ micro columns with 50 mM TEAB, 0.2% formic acid and 0.2% formic acid in 50% acetonitrile solution. Tryptic peptides were dried using a vacuum concentrator and reconstituted in 20 µl of 2% acetonitrile, 0.1% trifluoracetic acid.

## LC_MS/MS on Turbo-bio ID and Flag IP

Tryptic peptides were resuspended in 2% acetonitrile, 0.1% trifluoroacetic acid in LC-MS grade water and analysed by reverse-phase chromatography tandem mass spectrometry (LC-MS/MS) using an Ultimate 3000 nUHPLC connected to an Orbitrap Fusion Lumos (Thermo Fisher) as described previously (Liang et al, 2024; Yang et al, 2024). In brief, peptides were trapped onto a PepMapC18 trap column (300 µm × 5 mm, 5 µm particle size, Thermo Fisher) and separated on a 50 cm EasySpray column (75 µm, <2 µm particle size, ES803, Thermo Fisher) using a 60 min linear gradient from 2% to 35% buffer B (A: 5% DMSO, 0.1% formic acid; B: 5% DMSO, 0.1% formic acid in acetonitrile) at 250 nl/min flow rate. Data were acquired in the Orbitrap Fusion Lumos mass spectrometer in data-dependent mode (DDA) with the advance peak detection (APD) switched on. Full scans were acquired in the Orbitrap at 120 k resolution over a *m/z* range 400–1500, AGC target set to 4e5 and the S-lens RF of 30. MS2 scans were obtained in the Ion trap (rapid scan mode) with a Quad isolation window of 1.6, 40% AGC target and a maximum injection time of 35 ms, with HCD activation and 28% collision energy.

## Sample preparation for total proteome and phospho proteome profiling

Cells were lysed in RIPA buffer (50 mm Tris-HCl, 150 mM NaCl, 0.1 SDS, 0.5% Nadecoxylate, 1% NP40) supplemented with phosSTOP phosphatase inhibitors (Roche) and protease inhibitors. To remove the SDS, cell lysates (approximately 300 µg of protein material) were sequentially reduced with 5 mM DTT (final concentration) for 30 min at room temperature and alkylated with 20 mM IAA (final concentration) for 30 min at room temperature, before proteins were precipitated using Methanol: chloroform (Fielden et al, 2020). Protein pellets were resuspended with 8 M urea in 20 mM HEPES at pH 8 containing phosphatase inhibitors. Urea concentration was diluted down to <1 M with 20 m HEPES at pH 8 before adding Trypsin (ratio 1:50) for 16 h at 37 °C. Trypsin digestion was stopped by adding 1% TFA (final concentration). 10% of the tryptic digests were kept aside to perform a total proteome analysis. The rest of the tryptic digests were desalted using SOLA HRP cartridges with minor changes to manufacturer's instructions to suit the phospho-enrichment protocol. Briefly, cartridges were conditions with solvent B (70% acetonitrile, 0.1% TFA), washed with solvent A (0.1% TFA in H$_2$O), loaded with sample with 1:1 sample: solvent A ratio, washed with solvent A and eluted with 1 M glycolic acid in 50% acetonitrile, 5% TFA.

The phospho-enrichment step was performed using the predefine Agilent Bravo AssayMap liquid handler workflow for TiO$_2$ cartridges (AssayMap 5 µl Titanium Dioxide—TiO$_2$—cartridges, Cat Number G5496-60016, Agilent). The solutions used were 50% ACN, 5%NH3 in H$_2$O as prime and syringe wash solution, 50% ACN, 2% TFA, 1 M Glycolic acid in H$_2$0 as equilibration and cartridge wash and 15% acetonitrile, 5%NH$_3$ solution as elution buffer. Samples were loaded onto the cartridges at 2.5 µl/min. Eluted phosho-enriched peptides were dried down and resuspended in 0.1% Formic acid prior to analysis.

## LC-MS/MS of total proteome and phosphoproteome samples FBOX42 and PP4C

Phospho-enriched fractions were analysed by LC-MS/MS using the EvosepOne connected to a tims-TOF Pro mass spectrometer. Peptides were chromatographically separated on a 15 cm × 150 µm × 1.5 µm analytical column (EV1137, Evosep) using the 30 samples per day (spd) gradient method of an Evosep One LC system. Eluted peptides were directed for mass spectrometry (MS) analysis on a timsTOF Pro mass spectrometer (Bruker). Phosphopeptide MS data were acquired in data-dependent parallel accumulation serial fragmentation (ddaPASEF) mode. The ion mobility window was set to 0.60–1.60 Vs/cm$^2$, with an accumulation and ramp time of 100 ms. The mass range of MS and MS/MS scans was *m/z* 100–1700. MS/MS spectra were acquired in 10 PASEF ramps with a 4-frame overlap, giving a cycle time of 1.17 s. Ions were selected for PASEF MS/MS if they met an intensity threshold of 2500 and were sampled multiple times until a target intensity of 20,000 was reached. A polygon filter was used to exclude singly charged ions from MS/MS selection. Ions selected for fragmentation were isolated by the quadrupole and fragmented using an ion mobility-dependent collision energy that increased nonlinearly over the ion mobility range as follows: a collision energy (eV) of 20 at a 1/K0 of 0.60 Vs/cm$^2$, 22 at 0.70, 25 at 0.75, 30 at 0.80, 35 at 0.85,

40 at 0.90, 45 at 0.95, 50 at 1.00, 55 at 1.10, 60 at 1.20, 65 at 1.30, 70 at 1.40, 75 at 1.50, and 80 at 1.60. A dynamic exclusion time of 24 s was used.

The corresponding total proteomes were analysed using the Vanquish Neo UHPLC connected to Thermo Orbitrap Ascend mass spectrometer (all Thermo Fisher Scientific). The Vanquish Neo was operated in "Trap and Elute" mode using a PepMap Neo trap (5 µm, 300 µm × 5 mm; Thermo Fisher) with backflash and EASY-SPRAY PepMapNeo column (50 cm × 75 µm, 1500 bar; Thermo Fisher). Tryptic peptides were trapped and separated over a 75 min gradient, going from 3 to 20% B (0.1% FA in 80% ACN) in 40 min, to 35% in 20 min, up to 99% B in 1 min and then staying at 99% for 14 min. The flow rate was maintained at 300 nL/min throughout the gradient. MS data were acquired in Data Independent Acquisition (DIA) mode, with minor changes from our previously described method (Dellar et al, 2024; Muntel et al, 2019; O'Brien et al, 2023). Briefly, MS1 scans were collected in the Orbitrap mass analyser at a resolving power of 45 K at $m/z$ 200 over $m/z$ range of $m/z$ 350–1650. The MS1 normalised AGC was set at 125% (5e5 ions) with a maximum injection time of 91 ms and a RF lens at 30%. DIA MS2 scans were then acquired using the tMSn scan function at an Orbitrap resolution of 30 K over 40 scan windows of variable width, with a normalised AGC target of 1000%, maximum injection time set to auto and a 30% collision energy.

### Data analysis

FBOX42 turbo BioID and flag-IP mass spectrometry data were analysed using MaxQuant (v1.6.12.10 and v1.6.14, respectively). Briefly, files were searched against the UniProt-Swissprot human database (retrieved Mar 2020 containing 20365 sequences and Feb 2021 with 20381 sequences, respectively); using the in-build Andromeda data-search engine. Trypsin was selected as enzyme (up to 2 missed cleavages), carboamidomethylation (C) as fixed modification and Deamidated (NQ) and Oxidation (M) as variable modifications. Protein false discovery rate was set up at 1%. Data were quantified using the label free quantitation (LFQ) and the Intensity Based Absolute Quantification (iBAQ) parameter was enabled. Match between runs was not selected. Maxquant protein group output was further analysed using Perseus (1.6.2.2). Intensities were log 2 transformed, normalised by median subtraction before a 20% total valid number filter was applied and missing values were imputed (following the normal distribution). A two-sample student t-test was applied combined with a Permutation – FDR correction (5%).

FBOX42 and PP4C full proteome and phosphoproteome data were searched against the reviewed Human proteome (UniProtKB, downloaded 202202, 20386 sequences). Phospho-enriched samples were analysed in Fragpipe (v20.0) using the LFQ-phospho work-flow with the predefined settings, which included strict trypsin cleavage (2 missed cleavages), phosphorylation (STY), oxidation (M) and acetylation (N-terminus) as variable modifications and carbamidomethylation (C) as fixed modification. Match between runs was enabled. On the other hand, total proteomes were analysed using DIA-NN (v1.8.1) in library-free mode with default settings. Trypsin was selected as protease (1 missed cleavage), carbamidomethylation on Cysteines as fixed modification and Oxidation on methionine as variable (N-term M excision turn on).

Match between runs function was turn on. Search results were analysed in Perseus (v1.6.2.2). Data were log 2 transformed, normalised using the median subtraction before data were filtered based on valid number (3 in at least one biological group). Missing values were imputed following normal distribution (down shifted). Two sample student t-test combined with permutation FDR (set at 1%) was used. PCA, volcano plots and heat maps here used.

The mass spectrometry raw data included in this paper had been deposited to the Proteome eXchange Consortium via the PRIDE partner repository (Perez-Riverol et al, 2019). The dataset identifiers are as follows: FBXO42 Flag IP PXD069496, FBXO42 TurboID PXD069542 and full proteome and phosphoproteome data PXD069593.

### Total proteome analysis

Heatmap of the total proteomes of LN229 and FBXO42 K/O cells were analysed by hierarchical clustering using Euclidean distance and average for linkage method. The proteins from the clustering were used for pathway analysis using EnrichR.

### Phosphoproteomics analysis

Heatmap of the Phosphopeptides in LN229 and FBXO42 K/O cells and siRNA PP4C were analysed by hierarchical clustering using Euclidean distance and average for linkage method. The proteins cluster upregulated in siPP4C and downregulated in FBXO42 K/O cells were analysed using Kinase Enrichment Analysis version 3 (KAE3) to infer upstream kinases whose putative substrates are overrepresented in these differentially phosphorylated proteins. Pathway analysis was also completed on the FBXO42 K/O depleted phosphorylated proteins using EnrichR.

### Transfection, IP, and Wb

A total of 1.5 million HEK293T cells were seeded into each of the 10-cm petri dish 24 h before plasmid transfection. For each 10-cm dish, the polyethylenimine linear (PEI) transfection was performed by vigorously vortexing 5 µg of plasmid DNA with 15 µl of PEI (2.5 mg/ml) in 400 µl of plain DMEM (without FBS or antibiotics) for 15 s to mix, incubating the mixture for 15 min at room temperature, then adding it to cells cultured in 10-ml complete media in a dropwise fashion. PEI was purchased from Polysciences Inc. (23966). PEI stock (2.5 mg/ml) was made in 20 mM HEPES, 150 mM NaCl (pH 7.4), and filtered. Twenty-four hours after transfection, cells were washed twice with PBS and harvested. The cell pellets were stored at −80 °C or lysed directly for experiments.

Cell pellets harvested for IP or Wb (not aiming to detect DNA damage markers) were lysed in NP40 lysis buffer containing 50 mM Tris-HCl (pH 7.5), 150 mM NaCl, 1 mM EDTA, 5 mM MgCl$_2$, and 0.1% Nonidet P-40, supplemented with protease inhibitor cocktail (Sigma-Aldrich, P8340), 200 µM phenylmethylsulfonyl fluoride (PMSF; Santa Cruz Biotechnologies, sc-482875), and two phosphatase inhibitors, 20 mM β-glycerophosphate (Sigma-Aldrich, G5422) and 1 µM Okadaic acid (Cayman Chemical, 10011490-50 ug-CAY). After lysing on ice for 10 min, the insoluble fraction (mostly DNA and DNA bound proteins) was removed via centrifugation at 20,000 g at 4 °C for 15 min, and the supernatant was carefully transferred to new Eppendorf tubes without

disturbing the insoluble fraction. Protein concentration of the supernatant was measured using the modified Lowry assay (DC Protein Assay Kit, Bio-Rad, 5000111). The same amount of total protein was used for each IP (0.5 to 1 mg per pulldown in general) or direct Wb (10 to 20 µg per lane in general). Final samples were mixed with 4X Laemmli sample buffer (Invitrogen, NP0008) and boiled for ten minutes before being applied in SDS-PAGE.

For Flag IP, 10-µl Flag M2 beads (Sigma-Aldrich, A2220-5ML) or 15-µl HA beads (Sigma-Aldrich, E6779-1ML) were washed three times with lysis buffer and added to the cell lysates to incubate for 3 h on a roller at 4 °C. After incubation, the beads were collected by centrifugation and washed with inhibitor-containing lysis buffer five times before being mixed with 20–40 µl 1× Laemmli sample buffer (diluted from Invitrogen, NP0008) supplemented with β-mercaptoethanol, and boiled for 10 min at 95 °C. The boiled supernatant was used for Wb analysis.

For endogenous IP of PP4C components the lysates were pre-cleared with 5 µl protein A/G agarose beads for 1 h. The beads were removed by centrifugation and the supernatant incubated with the 1–3 µg of the relevant antibody overnight (Anti-PPP4C antibody STJ25096-100 or Anti-PP4R2 A300-838A). Next day, the supernatant was incubated with 10 µl of protein A/G agarose beads for 2–3 h. The beads were washed three times in lysis buffer and either immunoblotted or processed further for the dephosphorylation assay.

For experiments detecting DNA damage markers (which could be tightly chromatin bound), cell pellets were harvested, homogenized, and boiled directly in 2% SDS buffer [350 mM bis-tris (pH 6.8), 20% glycerol, and 2% SDS], then sonicated. Protein concentration was assessed using a BCA protein kit (Thermo Fisher Scientific, 23227). Cell lysate was prepared for Wb as indicated above and resolved in 4 to 12% gradient bis-tris gels, transferred onto nitrocellulose membrane (Amersham, 10600006) and immunoblotted. Wb results were visualized via X-ray film or iBright FL1500 Imaging System (Invitrogen, A44241).

## Cyclohexamide (CHX) chase

A total of 250,000 cells were seeded into each well of a six-well plate. Sixteen hours later, cells were cultured with CHX (50 µg/ml) for various durations as indicated in the figures, to block ribosomal protein synthesis for assessment of protein stability. Cells were collected via scraping and washed with PBS twice before being analysed by Wb for protein half-life estimation.

## In vivo ubiquitination assay

Two million HEK293T cells were seeded into each of the 10-cm dishes 24 h before being transfected with plasmids as indicated in each experiment (typically, for each 10-cm dish, 1 µg of substrate overexpression plasmid, 2 µg of E3 overexpression plasmid, or 3 µg of ubiquitin overexpression plasmid was co-transfected). Cells were cultured for another 24 h before being harvested. Four hours before the harvest, MG132 was added to a final concentration of 10 µM to block proteasomal degradation so that ubiquitination events were enriched. Cells were collected by scrapping and washed with PBS once, then thoroughly lysed and boiled in 300 µl of ubiquitin lysis buffer [2% SDS, 150 mM NaCl, and 10 mM Tris-HCl (pH 7.4)].

After cooling to room temperature, cell lysates were subjected to sonication until they lost their viscosity. Lysates were then boiled again and centrifuged at $17,000 \times g$ for 10 min. Twenty µl of supernatant was preserved as input for each sample. The rest of the supernatant was diluted 20 times with dilution buffer [10 mM Tris-HCl (pH 7.4), 150 mM NaCl, 2 mM EDTA, and 1% Triton X-100] and processed by IP of the substrate. After 16 h of incubation on a roller at 4 °C, beads were collected via centrifugation at 2000 rpm for 1 min and washed five times with 1 ml wash buffer [10 mM Tris-HCl (pH 7.4), 1 M NaCl, 1 mM EDTA, and 1% Nonidet P-40] before being mixed with 50 µl of 1X Laemmli sample buffer, boiled, and subjected to Wb.

## Ubiquitin binding entities (UBE) pulldown assay

Before harvesting cells, GST-UBA [ubiquitin-associated domain (UBA domain) of the UBQLN1 protein] was first conjugated to glutathione sepharose beads (Cytiva, GE17-0756-01) in ubiquitin binding entities (UBE) lysis buffer [19 mM $NaH_2PO_4$, 81 mM $Na_2HPO_4$ (pH 7.4), 1% Nonidet P-40, 2 mM EDTA, supplemented with protease inhibitor cocktail, PMSF, and phosphatase inhibitors as mentioned in the IP section], 50 mM N-ethylmaleimide (NEM; Sigma-Aldrich, E3876-5G), 5 mM 1,10-phenanthroline (Scientific Laboratory Supplies, CHE2730), and 50 µM PR-619 (ApexBio, A821) for at least 4 h at 4 °C on a roller. For each pulldown, 100 µg of recombinant GST-UBA was conjugated to 20 µl of washed glutathione beads. Cells were treated with 10 µM MG132 for 4 h before being harvested by scraping and centrifugation. Cell pellets were washed twice with PBS, then directly lysed in freshly prepared UBE lysis buffer. After incubation on ice for 10 min, lysates were subjected to centrifugation at 14,000 rpm at 4 °C for 15 min. After protein concentration was measured via Lowry assay using DC Protein assay kit (Bio-Rad, 5000111), the same amount of total protein was used for each pulldown (2 to 3 mg per pulldown). GST-UBA-conjugated beads were added to the lysate and incubated on a roller at 4 °C for overnight, then collected, washed using UBE lysis buffer, mixed with 1× Laemmli sample buffer, and boiled in the same way as described in the IP section. The supernatant was then used for Wb. Besides the protein of interest, total ubiquitin was probed as a loading control for UBE pulldown experiments.

## Analysis of previous published CRISPR screens

Publicly available CRISPR-Cas9 screening results were extracted from publications concerning sensitivity to ataxia telangiectasia mutated and RAD3-related (ATR) protein kinase inhibitors AZD6738 and VE821 (Hustedt et al, 2019; Wang et al, 2019) (Panel A); mitotic inhibitors BI-2536 and colchicine (Hundley et al, 2021) (Panel B); and X-ray irradiation (Yang et al, 2024) (Panel C). Genes were ranked according to lowest normalized gene-level Z-score (normZ) value for Panel A, in which a significance cut-off value was defined as $\leq -1.96$, lowest single guide RNA (sgRNA) median log2 fold change (Log2FC) for Panel B, in which a significance cut-off value was defined as $\leq -0.3$ as per publication methods, and lowest Z-ratio for Panel C in which the significance cut-off value was lowered to $\leq -1$ to reflect the use of focused CRISPR-Cas9 screening in place of genome wide.

## Generation of bio$^{GEF}$ UBnc cell lines and bio$^{GEF}$ UBnc+BIRA FBXO42 cell lines

Lentivirus expression construct TRIPZ-bio$^{GEF}$ UBnc puro (TRIPZ-bio$^{GEF}$-UBnc-PURO was a gift from Rosa Barrio & James Sutherland Addgene plasmid # 208044) were packaged in HEK293T cells by transfect with psPAX2, pMD2.G and pTAT (pcDNA1-Tat was a gift from Akitsu Hotta Addgene plasmid # 138478). Transfection media was removed after 12–16 h and replaced with fresh media. The lentivirus supernatants were collected after 48 h and filtered through 0.45 µM syringe filter. The lentivirus particles were used to transduce Flp-In™ T-REx™ 293 cells and antibiotic selection was performed with puromycin. To generate the bio$^{GEF}$ UBnc+BIRA FBXO42 cell line the bio$^{GEF}$ UBnc trex cell line were transfected with the pCDNA3–BIRA–FBXO42 plasmid and the pOG44 Flp-recombinase expression vector (Invitrogen, V600520) for coexpression of the Flp-recombinase using Lipofectamine 2000 transfection reagent (Invitrogen, 11668019). Two days after the transfection, cells were selected in hygromycin-containing medium (100 µg/ml) for 2 to 3 weeks. To validate the BIRA–FBXO42 expression, cells were cultured in media containing doxycycline (1.3 µg/ml) for 24 h to induce BIRA-FBXO42 expression before immunoblotting.

## BioE3/E-STUB

BIRA–FBXO42 *bio$^{GEF}$UBnc* Flp-In™ T-REx™ HEK293 cells grown in DMEM supplemented with 10% Tet FBS (PAN Biotech, P30-3602). One 15-cm dishes were used per experimental condition. When 15 cm reached 80% confluency, doxycycline (1.3 µg/ml) was added for 24 h to induce the expression of BIRA–FBXO42 and bio$^{GEF}$UBnc. Cells were supplemented with 50 µM biotin for 3 h before collection, pre-treatment with MG132 or MLN4924 was done 2 h before biotin was added. Cells were harvested by scraping and washed three times with phosphate-buffered saline (PBS). For streptavidin pulldown, cell pellets were thoroughly resuspended in lysis buffer [8 M Urea (in 1xPBS) 1% SDS, 50 µM NEM and Protease inhibitor cocktail (Sigma-Aldrich, P8340)] and incubated on ice for 15 min and sonicated 3 cycles of 30 s on 30 s off. Insoluble material was removed by centrifugation. Cleared lysates were incubated on a rotating wheel at 4 °C with 50-µl pre-equilibrated Strep-Tactin® Superflow® high-capacity resin (IBA, 2-1208-002) overnight. The beads were washed 3 times in lysis buffer and 3 times in RIPA buffer. Beads were then resuspended in 2x Laemmli Sample Buffer (Biorad#1610747) and boiled at 95 °C for 5 min. Samples were then analysed by western blot.

## Flow cytometry

EdU (5-ethynyl-2'-deoxyuridine) incorporation in LN229 cells was measured using the Click-iT™ EdU Cell Proliferation Kit for Imaging. Exponentially growing LN229 (parental or FBXO42 knockout) cells were incubated with media supplemented with 10 µM Edu for 45 min after which they were quickly washed with PBS, trypsinysed and fixed in ice cold 70% ethanol and stored at −20 °C (for at least 2 h). Fixed cells were washed with PBS and permeabilized using PBS-Triton™ X-100 (0.5%). After a brief incubation in 3% BSA in PBS/Tween® 20 (0.05%), Edu incorporated cells were labelled with AlexaFluor® 647 according to the manufacturer's instructions. Cells were subsequently washed with PBS-Tween® 20 (0.5%) and incubated in DAPI (final concentration 10 µg/ml) in the presence of ribonuclease A (final concentration 100 µg/ml) for 30 min at room temperature in the dark prior to analysis by flow cytometry in PBS supplemented with BSA (0.1% w/v) and EDTA (0.5 mM). Cell cycle distribution was measured on a Beckman Coulter CytoFLEX S Flow Cytometer using the 638 nm laser (660/10 bandpass filter) for detection of AlexaFluor® 647 and the 405 nm laser (450/45 bandpass filter) for DAPI. Gates were set to record over 25,000 (typically 30,000) single cell events in three experimental repeats. Analysis of flow cytometry data was carried out using FlowJo™ v11 Software (BD Life Sciences) and visualised using GraphPad Prism version 10.6.1 for Windows.

## Dephosphorylation assay

Immediately after the completion of the endogenous IP the beads were washed three times in 50 mM TRIS pH 7.4 with 150 mM NaCl. The phosphorylated peptide KRpTIRR was resuspended in 50 mM Tris and used in the assay at a final concentration of 230 µM. The reaction was allowed to proceed for 1 h and the release phosphate was detected using the Biomol® green reagent, with absorbance measured at 620 nM.

## Data availability

The mass spec data from this publication have been deposited in the PRIDE database and can be accessed as follows: FBXO42_-FlagIP PXD069496, FBOX42_TurboBioID PXD069542 and FBOX42 K/O and PP4C K/O Total and Phosphoproteomics PXD069593.

The source data of this paper are collected in the following database record: biostudies:S-SCDT-10_1038-S44318-025-00675-y.

## Peer review information

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

## Acknowledgements

This study was supported by Medical Research Council (MRC) grant MR/X006980/1 to VD and a Cancer Research UK (CRUK) grant DRCNPG May21\100002 to VD. Further support is provided by participation in the strategic network *Glasgow Radiation Research Centre of Excellence* RRCOER-Jun24/100003. We acknowledge further support by the John Fell Fund (133/075), Wellcome Trust (097813/Z/11/Z), and EPSRC (EP/N034295/1) to BMK. RF, IV, and BMK are supported by the Chinese Academy of Medical Sciences (CAMS) Innovation Fund for Medical Science (CIFMS), China (grant number: 2018-I2M-2-002). We would like to thank Elisabeth Freyer and Colin Stok for help with the Flow cytometry analysis.

## Author contributions

**Hongbin Yang**: Conceptualization; Data curation; Formal analysis; Supervision; Validation; Investigation; Visualization; Methodology; Writing—original draft; Project administration; Writing—review and editing. **Paul Smith**: Conceptualization; Resources; Data curation; Software; Formal analysis; Supervision; Validation; Investigation; Visualization; Methodology; Writing—

original draft; Project administration; Writing—review and editing. **Yingying Ma**: Conceptualization; Data curation; Formal analysis; Validation; Investigation. **Emily Southworth**: Conceptualization; Data curation; Formal analysis. **Varun Gopala Krishna**: Conceptualization; Data curation; Formal analysis; Writing-original draft. **Beatrice Salerno**: Conceptualization; Data curation; Formal analysis. **Joseph Rowland**: Conceptualization; Data curation; Formal analysis. **Alexander E P Loftus**: Data curation; Formal analysis; Visualization; Writing—review and editing. **Domenico Grieco**: Conceptualization. **Iolanda Vendrell**: Conceptualization; Data curation; Software; Formal analysis; Methodology. **Roman Fischer**: Conceptualization; Data curation; Formal analysis; Methodology. **Benedikt M Kessler**: Conceptualization; Formal analysis; Methodology. **Vincenzo D'Angiolella**: Conceptualization; Resources; Data curation; Formal analysis; Supervision; Funding acquisition; Validation; Investigation; Visualization; Methodology; Writing—original draft; Project administration; Writing—review and editing.

Source data underlying figure panels in this paper may have individual authorship assigned. Where available, figure panel/source data authorship is listed in the following database record: biostudies:S-SCDT-10_1038-S44318-025-00675-y.

## Disclosure and competing interests statement

The authors declare no competing interests.

# Expanded View Figures

**Figure EV1.  FBXO42 loss sensitises cells to ATR kinase inhibitors and microtubule depolymerising agents.**

(A) CRISPR chemical-genetic screening results denoting vulnerabilities in response to ATR kinase inhibitors in RPE1 hTERT p53-/- and MCF10A cells (Hustedt et al, 2019; Wang et al, 2019), using a normZ significance cut-off of $\leq -1.96$. FBXO42 is highlighted. (B) CRISPR chemical-genetic screening results which details vulnerabilities in response to tubulin inhibitor colchicine in HAP1 cells (Hundley et al, 2021), using a median sgRNA log2 fold change significance cut-off of $\leq -0.3$. FBXO42 is highlighted. (C) Dose response curves for ATR inhibitor VE821 to test the effect of FBXO42 K/O on cell survival in LN229 cell line. Data presented as mean $+/-$ SD; $n = 3$ biological replicates. (D) Dose response curves for ATR inhibitor VE821 to test the effect of FBXO42 K/O on cell survival in H4 cell line. Data presented as mean $+/-$ SD; $n = 3$ biological replicates. (E) Dose response curves for Indibulin to test the effect of FBXO42 K/O on cell survival in LN229 cell line. Data presented as mean $+/-$ SD; $n = 3$ biological replicates. (F) Dose response curves for Indibulin to test the effect of FBXO42 K/O on cell survival in H4 cell line. Data presented as mean $+/-$ SD; $n = 3$ biological replicates. (G) Colony formation assay in FBXO42 K/O cells. H4 parental cells or H4 FBXO42 K/O cells were seeded for colony formation assay and challenged with the indicated dose of IR. Seven days after IR, cells were stained with crystal violet and counted. Data presented as mean $+/-$ SD; $n = 3$ biological replicates. Two-tailed unpaired t test was performed as statistical analysis. Exact P values < 0.0001. **$P \leq 0.01$; ***$P \leq 0.001$; ****$P \leq 0.0001$. (H) Immunoblotting after IR treatment and 3-h recovery in LN229 parental and FBXO42 K/O cells. DNA damage markers detected: pCHK1 S296 (cell cycle checkpoint), pRPA32 S4/8 (ssDNA at DSB), pRPA32 S33 (single-strand breaks), and γH2Ax (DNA DSBs).

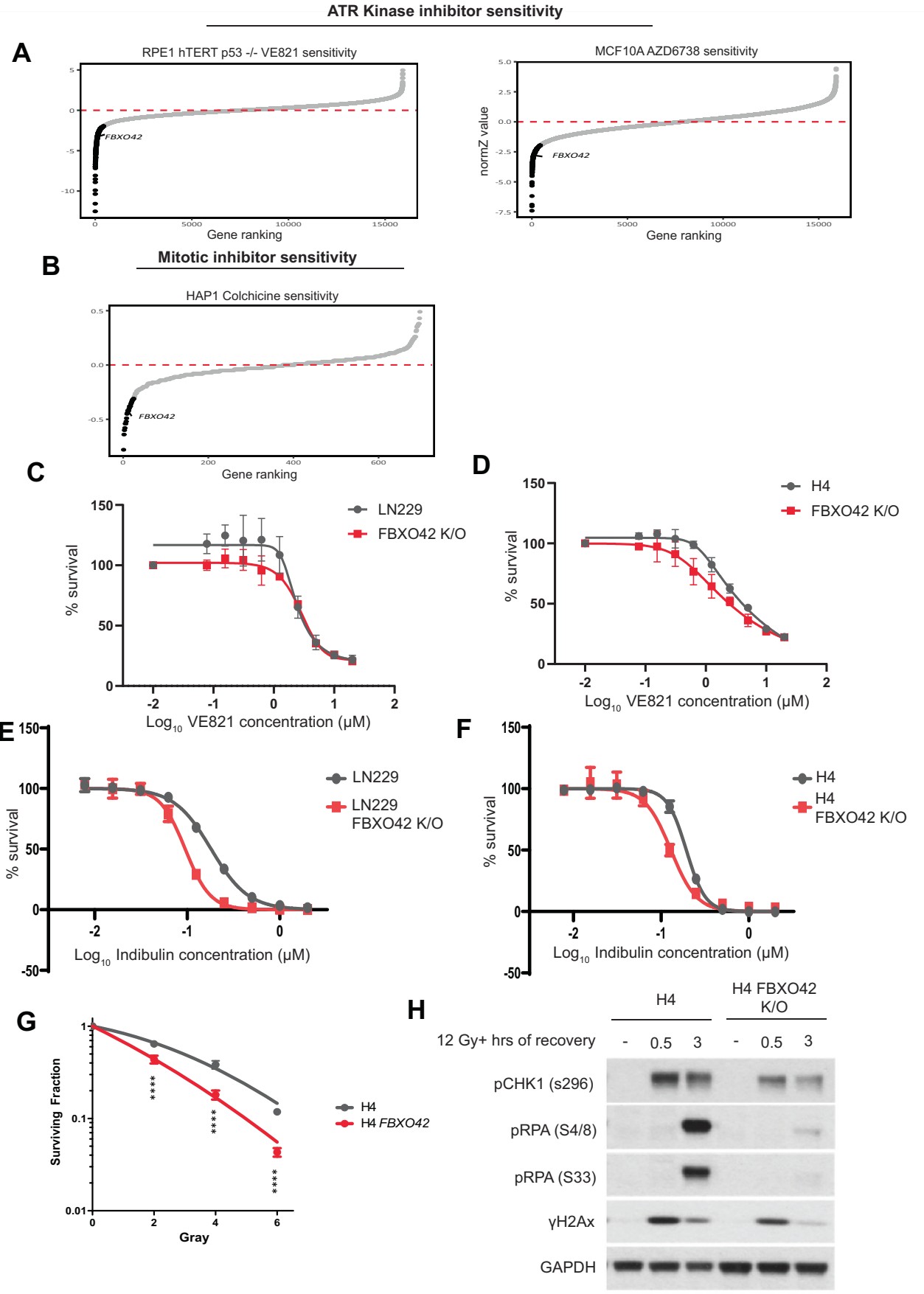

**ATR Kinase inhibitor sensitivity**

**A**

RPE1 hTERT p53 -/- VE821 sensitivity

MCF10A AZD6738 sensitivity

**B** **Mitotic inhibitor sensitivity**

HAP1 Colchicine sensitivity

**C**

**D**

**E**

**F**

**G**

**H**

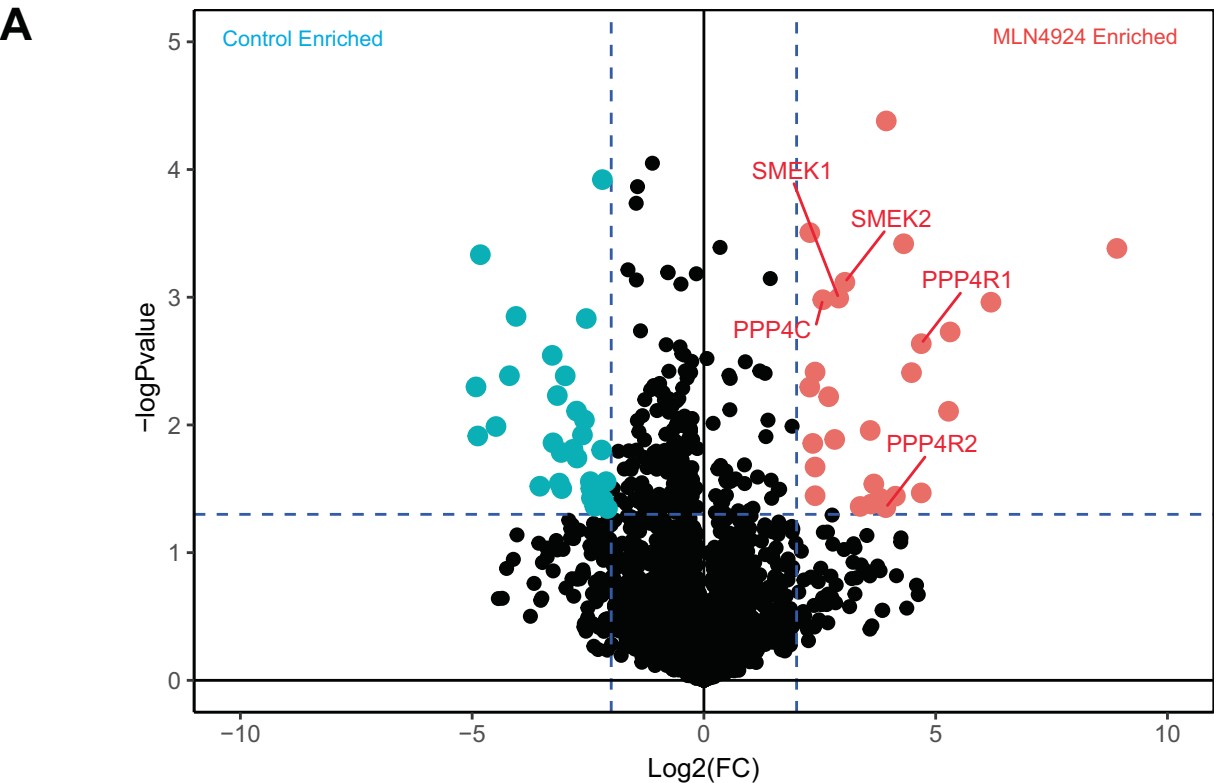

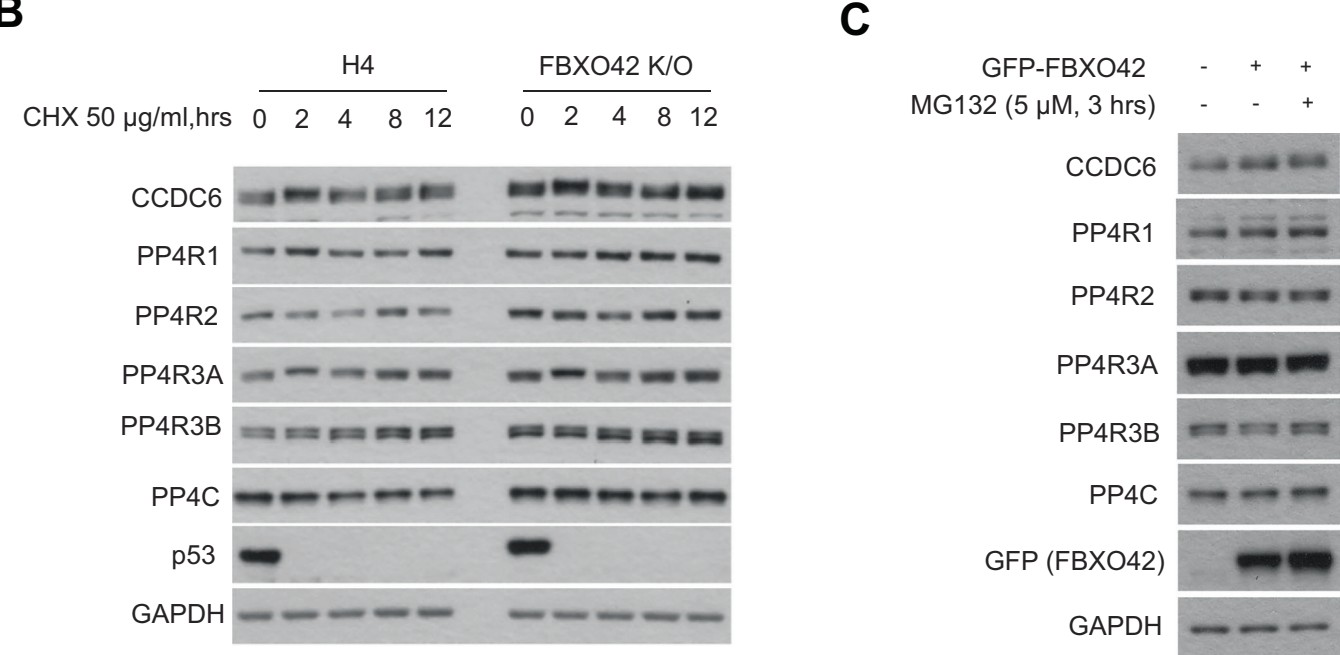

◀ **Figure EV2. FBXO42 interacts with the PP4 complex.**

(A) Volcano plot representing differentially enriched proteins (+/− MLN4924) generated using TurboID–FBXO42 followed by mass spectrometry. Interacting proteins were isolated with streptavidin after labelling for 1 h with biotin. Proteins considered significant have a *p*-value greater or equal to 0.05 and log-fold change >2; two-tailed t-test, Benjamini–Hochberg corrected. In red are proteins enriched upon MLN4924, in blue enriched in untreated samples. (B) Immunoblotting of FBXO42 interactors after treating H4 parental or FBXO42 K/O cells with cycloheximide (CHX) for the indicated times (hrs = hours). (C) Immunoblotting of PP4 subunits after expression of GFP-FBXO42 in HEK293T with or without proteasomal inhibition using MG132 (5 μM) as indicated. Source data are available online for this figure.

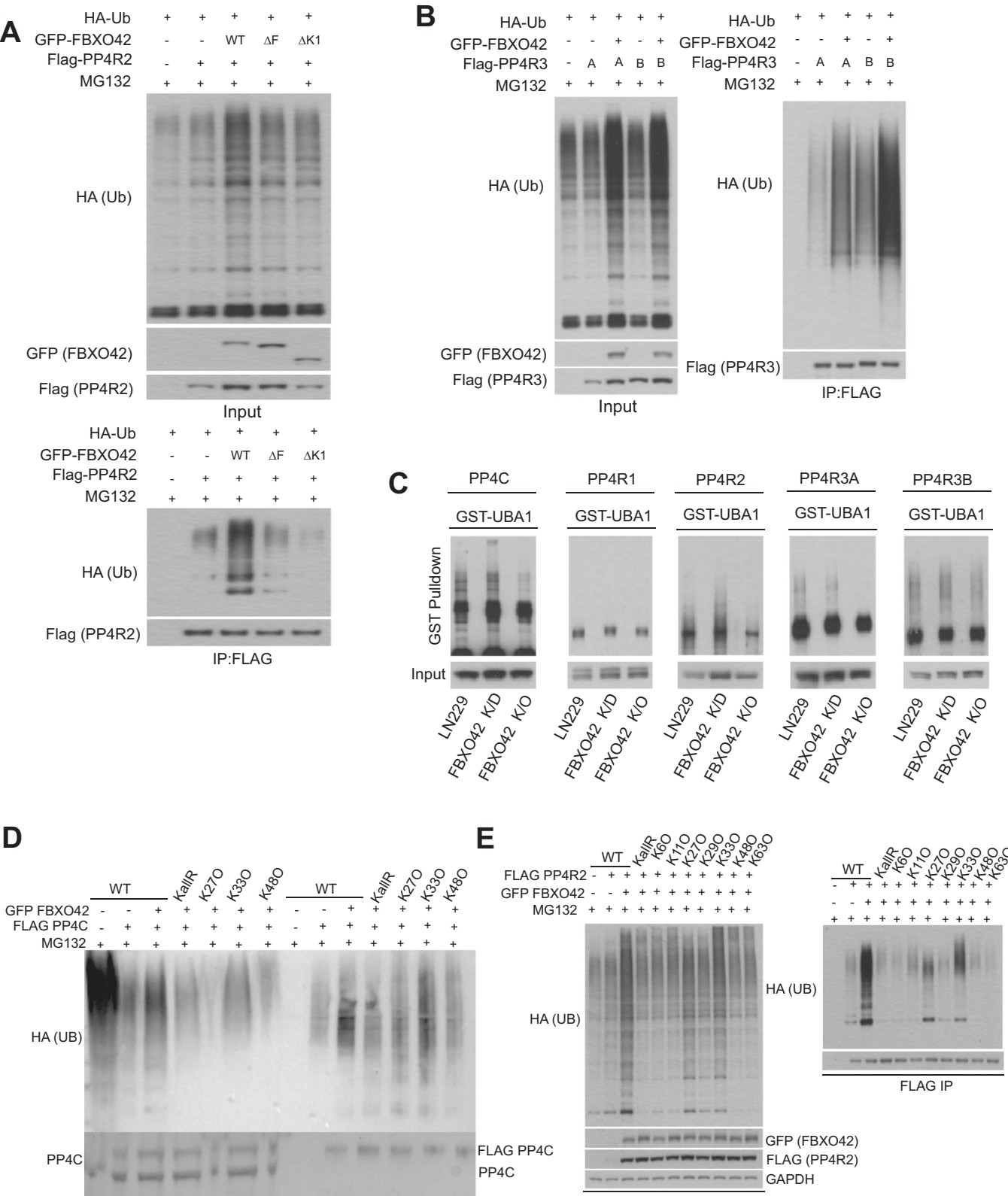

◄  **Figure EV3.  PP4 subunits are ubiquitinated by FBXO42.**

(**A**) Immunoblotting after expression of HA-ubiquitin, Flag-PP4R2 and either GFP-FBXO42 WT, F-box (F) mutant and Kelch (K1) mutant in HEK293T. Flag-PP4R2 was isolated via Flag agarose beads pulldown under denatured conditions before immunoblotting. Input samples are in the top panel. (**B**) Immunoblotting after expression of GFP-FBXO42, HA-ubiquitin and either Flag-PP4R3A or Flag-PP4R3B in HEK293T. Flag-PP4R3A or Flag-PP4R3B (A or B) was isolated via Flag agarose beads pulldown under denaturing conditions before immunoblotting. Input samples are in the left panel. (**C**) Immunoblotting after isolation of endogenous ubiquitinated proteins using recombinant GST-tagged UBA domain of UBIQLN protein from LN229 parental, FBXO42 K/D (partial knockout of FBXO42) or FBXO42 K/O cells. Input samples before immunoprecipitations are indicated in the bottom panel. (**D**) Immunoblotting after expression of GFP-FBXO42, HA-ubiquitin WT or one of the ubiquitin K to R mutants as indicated and Flag-PP4C in HEK293T. Flag-PP4C was isolated via Flag agarose beads pulldown under denaturing conditions before immunoblotting. (**E**) Immunoblotting after expression of GFP-FBXO42, HA-ubiquitin WT or one of the ubiquitin K to R mutants as indicated and Flag-PP4R2 in HEK293T. Flag-PP4R2 was isolated via Flag agarose beads pulldown under denaturing conditions before immunoblotting. Source data are available online for this figure.

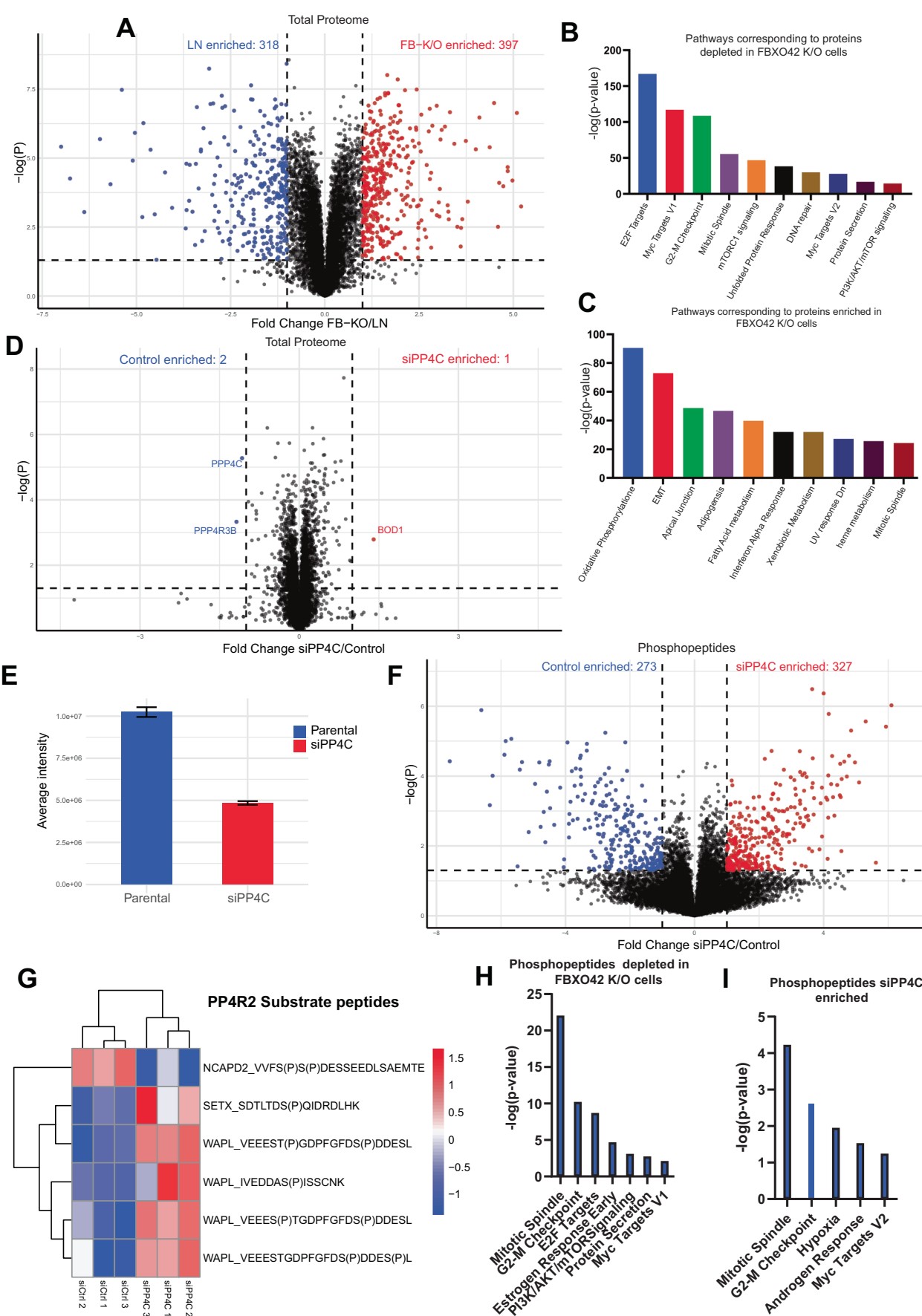

◀ **Figure EV4. FBXO42 K/O mediated changes in total proteome and phosphoproteome.**

(A) Volcano plot depicting total proteomes of LN229 parental cells compared to FBXO42 K/O cells. Proteins were considered significant if the *p*-value is greater or equal to 0.05 and fold change is greater or equal to 2; two-tailed t-test, Benjamini–Hochberg corrected. Significant proteins are highlighted in red or blue. (B) Gene enrichment analysis of proteins accumulating in parental cells compared to FBXO42 K/O. The MSigDB hallmarks 2020 enrichment tool on EnrichR is used. (C) Gene enrichment analysis of proteins depleted in LN229 parental cell lines compared to FBXO42 K/O. The MSigDB hallmarks 2020 enrichment tool on EnrichR is used. (D) Volcano plot depicting total proteomes of LN229 cells treated with siPP4C or siCtrl. Proteins were considered significant if the *p*-value is greater or equal to 0.05 and fold change is greater or equal to 2; two-tailed t-test, Benjamini–Hochberg corrected. Significant proteins are highlighted in red or blue. (E) Relative intensity of PP4C protein derived from the LC/MS datasets presented in (D). Error bars represent SDs of three biological replicates. (F) Volcano plot depicting differentially enriched phosphorylated motifs/phosphopeptides of LN229 cells treated with siPP4C or siCtrl. Peptides were considered significant if the *p*-value is greater or equal to 0.05 and fold change is greater or equal to 2; two-tailed t-test, Benjamini–Hochberg corrected. Significant proteins are highlighted in red or blue. (G) Heatmap of phosphopeptides in LN229 cells treated with siRNA against PP4C compared to LN229 treated with siRNA control. The peptides from proteins considered to be substrates of the PP4R2 subcomplex previously annotated by (Ueki et al, 2019) and considered be significant *P*-value 0.05 and fold change greater or equal to 1. (H) Gene enrichment analysis of proteins who's corresponding phosphopeptides were dephosphorylated/reduced in LN229 FBXO42 K/O cell lines. The MSigDB hallmarks 2020 enrichment tool is used on EnrichR. (I) Gene enrichment analysis of proteins who's corresponding phosphopeptides were phosphorylated/induced in LN229 cells treated with siRNA against PP4C. The MSigDB hallmarks 2020 enrichment tool is used on EnrichR.

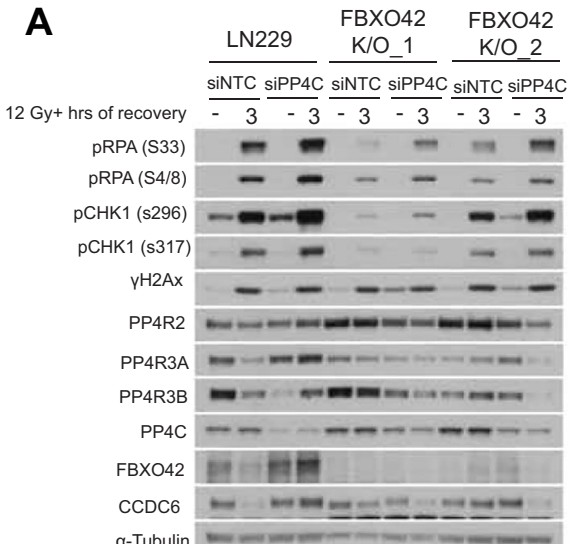

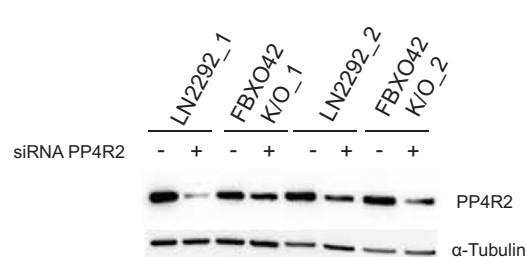

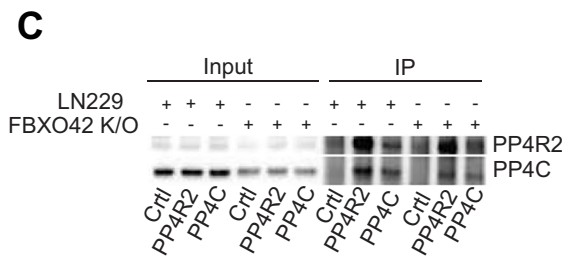

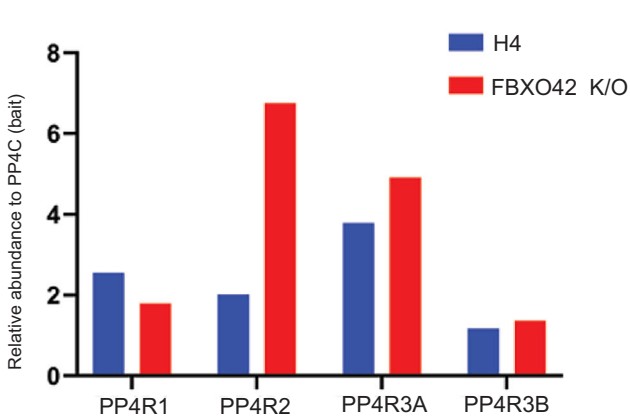

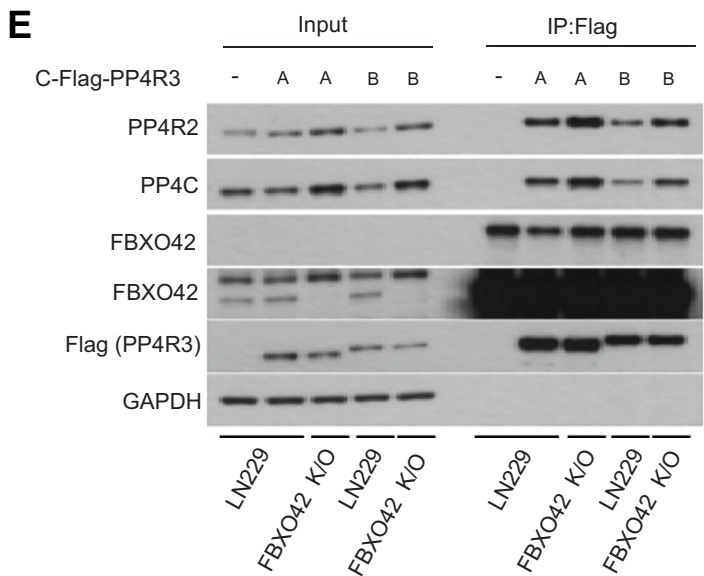

◀ **Figure EV5.  FBXO42 controls the activity of PP4 phosphatase and subunit assembly.**

(A) Immunoblotting of DNA damage markers and PP4 complex subunits in LN299 parental and FBXO42 K/O cell lines (clones 1 and 2) after IR treatment and 3-hours recovery with or without siRNA knockdown of PP4C. DNA damage markers detected: pRPA32 S33 (single-strand breaks), pRPA32 S4/8 (ssDNA at DSB), pCHK1 S296 (cell cycle checkpoint), pCHK1 S317 and γH2Ax (DNA DSBs). (B) Immunoblotting of PP4R2 levels after treatment with siRNA targeting PP4R2 or siRNA control. (C) Immunoblotting of PP4C and PP4R2 after immunoprecipitation with the indicated antibodies. Related to the activity of PP4 measured in Fig. 5C. (D) Relative abundance of PP4R1, PP4R2, PP4R3A and PP4R3B to PP4C (bait) determined by comparing LFQ Intensity in H4 parental cell lines compared to H4 FBXO42 K/O after immunoprecipitation of Flag PP4C and LC/MS. (E) Immunoblotting of PP4 subunits, as indicated, after expression of C-Flag-PP4R3A or Flag-PP4R3B in LN229 parental or FBXO42 K/O cell lines. Flag-PP4R3A/3B was isolated via Flag agarose beads pulldown before immunoblotting. Input samples are indicated on the left. Source data are available online for this figure.

