## [Peer Review File · The EMBO Journal]

Pervasive phenotypic effects of FBXO42 promoted by regulation of PP4 phosphatase

Hongbin Yang, Paul Smith, Yingying Ma, Emily Southworth, Varun Gopala Krishna, Beatrice Salerno, Joseph Rowland, Alexander Loftus, Domenico Grieco, Iolanda Vendrell, Roman Fischer, Benedikt Kessler, and Vincenzo D'Angiolella

Corresponding author(s): Vincenzo D'Angiolella (vdangio@ed.ac.uk)

Review Timeline:

Submission Date:	20th May 25
Editorial Decision:	27th Jun 25
Revision Received:	3rd Nov 25
Editorial Decision:	28th Nov 25
Revision Received:	9th Dec 25
Accepted:	12th Dec 25

Editor: Hartmut Vodermaier

Transaction Report:

Prof. Vincenzo D'Angiolella
University of Edinburgh
Edinburgh Cancer Research Centre - Institute of Genetics and Cancer
2XU, Crewe Rd S, Edinburgh
Edinburgh
United Kingdom

27th Jun 2025

Re: EMBOJ-2025-121417
Pervasive phenotypic effects of FBXO42 promoted by regulation of PP4 phosphatase

Dear Vincenzo,

Thank you again for submitting your study on FBXO42-PP4 regulatory interplay to The EMBO Journal. It has now been seen by three expert referees, whose comments are copied below. As you will see, referees 1 and 2 are both overall supportive, while referee 3 would have wanted more follow-up investigation in pathophysiological and in vivo contexts. With this aspect in my view going somewhat beyond the scope of the already comprehensive characterization, we conclude that we would like to pursue a revised version of this manuscript further for publication, provided that you should be able to address the various other, specific points raised in the three reports. In particular, I feel that it would be important assess effects on PP4 activity and test the possible ubiquitination roles of PP4C C-terminal lysines (ref 1), to look at cell-cycle distribution as well as epistasis/rescue of cellular phenotypes (ref 2), and to follow-up on the basis of altered PP4C protein levels (ref 3).

Please be reminded that our single-major-revision-round policy makes it important to diligently respond to each referee point at the time of resubmission; therefore, please do not hesitate to contact me early on in case you would like to clarify/discuss any of the referees' points or plans for answering. We would also be open to extending the revision deadline if that should be helpful. Our scooping protection policy means that competing manuscripts published while your work is under revision will not have a negative effect on our final decision.

Detailed information on preparing, formatting and uploading a revised manuscript can be found below and in our Guide to Authors, and adhering to them as closely as possible shall greatly facilitate editorial processing upon resubmission. Thank you again for the opportunity to consider this work for The EMBO Journal, and I look forward to your revision in due time.

Yours sincerely,

Hartmut Vodermaier

3) Revised manuscript text (including main tables, and figure legends for main and EV figures) has to be submitted as editable

text file (e.g., .docx format). We encourage highlighting of changes (e.g., via text color) for the referees' reference.

4) Each main and each Expanded View (EV) figure should be uploaded as individual production-quality files (preferably in .eps, .tif, .jpg formats). For suggestions on figure preparation/layout, please refer to our Figure Preparation Guidelines:

8) Please note that supplementary information at EMBO Press has been superseded by the 'Expanded View' for inclusion of additional figures, tables, movies or datasets; with up to five EV Figures being typeset and directly accessible in the HTML version of the article. For details and guidance, please refer to:

embopress.org/page/journal/14602075/authorguide#expandedview

9) To facilitate reproducibility and cross-laboratory adoption of methodologies, please structure the Materials & Methods section as outlined in our guide to authors, including a completed Reagents and Tools Table that can be downloaded from our author guidelines as well (<https://www.embopress.org/page/journal/14602075/authorguide#structuredmethods>).

10) Digital image enhancement is acceptable practice, as long as it accurately represents the original data and conforms to community standards. If a figure has been subjected to significant electronic manipulation, this must be clearly noted in the figure legend and/or the 'Materials and Methods' section. The editors reserve the right to request original versions of figures and the original images that were used to assemble the figure. Finally, we generally encourage uploading of numerical as well as gel/blot image source data; for details see: embopress.org/page/journal/14602075/authorguide#sourcedata

Further information is available in our Guide For Authors:

In the interest of ensuring the conceptual advance provided by the work, we recommend submitting a revision within 3 months (25th Sep 2025). Please discuss the revision progress ahead of this time with the editor if you require more time to complete the revisions. Use the link below to submit your revision:

Link Not Available

Referee #1:

Regulation of cellular processes by PTMs such as phosphorylation and ubiquitination is key for cellular fitness and disruption of PTM regulators can confer synthetic lethality to chemotherapeutic agents. The entry point to this work is the general interest of the lab in ubiquitin regulated processes and DNA repair. The authors investigate published genome wide synthetic lethality CRISPR screens for ATR inhibitors which uncovered FBXO42 as a possible synthetic lethal gene. FBXO42 was also picked up in as a synthetic lethal interaction in cells treated with IR (work from this lab). FBXO42 is an SCF adaptor with an unusual kelch domain that is poorly characterised and thus the authors wanted to understand the role of FBXO42 in controlling DNA repair. First, they validate the results from the genome wide screens by generating FBXO42 KO glioblastoma cell lines and show sensitivity to ATRi and IR. They observe that in these cell lines DNA repair is compromised based on phosphor antibodies for Chk1 and RPA.

To further investigate this the authors, look for FBXO42 substrates taking proteomic approaches and identifying interactors in the presence of a NEDD8 inhibitor and IR. They observe multiple components of the PP4 complex and the PP4 interactor CCDC6 in cells treated with the NEDD8 inhibitor suggesting that these might be targeted for ubiquitination by FBXO42. Consistent with this the deletion of the F-box domain resulted in robust binding to PP4 components which was dependent on the kelch domain. Analysis of PP4 complex subunit stability in FBXO42 null cells suggests limited effects arguing for none-

degradative ubiquitination. A slight increase in PPP4C and PPP4R2 protein levels are observed in FBXO42 null cells. Elegant experiments suggests that PPP4C, the catalytic subunit, is the likely target for ubiquitination while the other subunits of the complex are less likely targets. Phosphoproteomic analysis of WT vs FBXO42 null cells and correlation with known PP4 substrates suggests that PP4 is more active in FBXO42 null cells arguing that ubiquitination of PPP4C has a negative effect on PP4 holoenzymes. The authors confirm this at two levels: firstly the decreased phosphorylation on Chk1 and RPA in FBXO42 null cells can be partly rescued by PPP4C depletion and secondly in the absence of FBXO42 there seems to be more binding of PPP4R2 to PPP4C. Collectively the authors conclude that FBXO42 ubiquitinates PPP4C to prevent specific PP4 holoenzymes from forming hereby restraining phosphatase activity.

Overall, this is a nicely conducted study that has a coherent flow and logic behind the different experiments. I therefore think it is a potential good candidate for EMBO J but I have a few points for experiments to improve the work.

- 1) At no point do the authors actually look at the activity of PP4 in the presence/absence of FBXO42. I would recommend to IP PP4 complexes and conduct phosphatase assays on the beads with a peptide substrate to at least confirm that FBXO42 is a negative regulator of PP4. As they also observe PP2A components in the FBXO42 MS this should be included for completeness/specificity.
- 2) Is localisation of PP4 components affected by FBXO42 loss?
- 3) Did the authors manage to map the ubiquitination sites on PPP4C? Uniquely PPP4C has two lysine residues at its very C-terminus (one has been reported to be ubiquitinated) and since the C-terminus of the PPP family is often a critical point of regulation it could be that these are the critical residues. Maybe this could be easily tested with K to R mutations.
- 4) An expansion of how FBXO42 recognises PP4 complexes and the role of CCDC6 in this would be interesting but not strictly needed.
- 5) Citations: please also cite Lipinszki et al 2015 as they were the first to describe the PP4 motif we just expanded on this work.

Referee #2:

Summary

The manuscript from Yang and colleagues examines the functional and mechanistic role of FBXO42, a poorly studied member of the F-box family of E3 ligase substrate receptors. Despite recent evidence suggesting an important function in cell proliferation and genome stability, the mechanism underlying this remained unknown. They show here that FBXO42 function is dependent on protein phosphatase PP4, which is ubiquitinated in an FBXO42 dependent manner. The author's data suggests an interesting, and someone counterintuitive possibility, which is that ubiquitination disrupts the assembly of the complex, and that in the absence of FBXO42, PP4 activity is unleashed, leading to widespread changes in protein phosphorylation and likely abundance, and that these collectively contribute to cellular phenotypes involved in cell cycle and DNA damage. Overall, this is a very interesting study and would be of interest to the readership of EMBO. There are some points generally minor points that could be clarified or experimentally addressed that would strengthen the conclusions of the study and make it clearer to read.

Major points

- Does FBXO42 knockout significantly alter the overall cell cycle distribution? This could be relevant to the interpretation of phosphoproteomics and proteomics studies. A simple cell cycle analysis would be a helpful addition and could be done using the already generated KO cell lines.
- Could some cellular phenotypes attributed to FBXO42 loss be rescued by depletion of PP4R2 or, maybe more difficult, PP4C?
- Overall, the phosphorylation and proteomic analysis could be better described. The number of proteins and sites identified is unclear, as are the broad details for how the MS was done (e.g., DIA v TMT, fractionation, etc.). How was quantification performed? How many proteins or phosphorylation sites were identified?
- It is not clear what is being depicted in Figure 4A. Are these phosphorylation sites? If so, was only one site changed in each of these substrates? If it is a specific phosphorylation site, what are those sites and were they the same sites identified in the previously done PP4 proteomics analysis? Relatedly, why do the authors refer to 14 proteins in the text, but show fewer in this figure?
- What is the source of proteomics data analyzing PP4 knockdown by siRNA? Is this published, or is this a dataset generated here? Similar to the comment above, I believe this was done as part of this study, but the details of the experiment are not mentioned, number of sites identified, an experimental detail, etc. Importantly, how does this dataset overlap with the FBXO42 KO dataset (not just the putative kinases) and the previous publication on PP4 substrates?
- It would be helpful to show the data in Figure 4E in some sort of graph (bar graph?). As is, it is an unusual way to display these data and it is not quickly apparent what it is.

Minor points

- BI2536 likely can inhibit other PLKs (mentioned in intro)
- The analysis in Figure 1 is compelling that FBXO42 is recurrently identified as being involved in the response to ATR inhibition and mitosis. It would be interesting to show an overlap from the strong hits of these screens. How many genes overlap between these studies? Is FBXO42 the only E3 or F-box protein that scores in this analysis. A simple Venn diagram showing the overlapping hits would be interesting and providing these data in a supplemental table could be useful for others in the field interested in similar pathways.

- Was the Enrichr analysis done on proteins that were altered by FBXO42 KO or on phosphoproteins altered? Similar to the comments above, the proteomics analysis and its discussion are in need of additional clarity throughout.
- Clarify on line 327-328 that the upregulation referred to is phosphorylation.

Referee #3:

In this manuscript, the authors demonstrate that FBXO42 negatively regulates PP4C activity. FBXO42 depletion leads to DDR deficiency. Substrate screening identified PP4C as a target of FBXO42 through ubiquitination, which does not promote proteasomal degradation but rather suppresses PP4C activity. They employed a comprehensive biochemical approach to characterize the FBXO42-PP4C pathway. However, the downstream consequence of ubiquitin-mediated regulation of PP4C in cancer remains unexplored.

Major Concerns

1. FBXO42 is reported to be a critical regulator of glioblastoma stem-like cells, as well as in melanoma and other cancer types. This study would benefit from further analysis of the effects of FBXO42-mediated PP4C regulation in cancer, using in vitro models, mouse models, and/or clinical tumor specimens.
2. The PP4C protein level is increased in FBXO42 KO cells, as shown in Figures 2D, 2E, and 4E. Is this due to post-transcriptional regulation or prolonged protein half-life? Although the authors excluded the possibility of increased protein stability, they should verify that their CHX assay was properly conducted by evaluating multiple cell lines and testing a range of CHX concentrations. A pulse-chase experiment is another widely accepted approach to confirm changes in protein stability and should be considered.

Minor Concerns

1. CCDC6 is presented as upregulated in both FBXO42-KO and parental cell groups in Figure 4A.
2. It would be beneficial to determine which lysine linkage is dominant in PP4C modification.

CANCER
RESEARCH
UK

Scotland
Centre

Professor Vincenzo D'Angiolella
Charles and Ethel Barr Chair Professor of Cancer Research
Edinburgh Cancer Research
Cancer Research UK Scotland Centre,
The Institute of Genetics and Cancer
University of Edinburgh
Crewe Road South
Edinburgh
EH4 2XU
Email: vdangio@ed.ac.uk

Dear Hartmut,

We have now reviewed our manuscript in line with your comments and the reviewer's feedback. We hope that the new updated version of the manuscript is a suitable candidate for *EMBO J*. Please find attached a point-to-point rebuttal:

Referee #1:

We would like to thank the reviewer for the overall positive view on our work. In reply to the reviewer questions, we have now conducted further experiments to address the raised points.

1) At no point do the authors actually look at the activity of PP4 in the presence/absence of FBXO42. I would recommend to IP PP4 complexes and conduct phosphatase assays on the beads with a peptide substrate to at least confirm that FBXO42 is a negative regulator of PP4.

To test the activity of the PP4 complex we have now measured the ability of the PP4C/PP4R2 complex, isolated from parental cells and cells FBXO42 K/O, to dephosphorylate a canonical peptide substrate KR(p)SIRR. The experiment presented in the updated Figure 5C shows that the activity of the PP4C complex is significantly increased in cells lacking FBXO42. The levels of the baits (FigureEV5C) were comparable in the experiment suggesting that FBXO42 controls PP4 activity.

As they also observe PP2A components in the FBXO42 MS this should be included for completeness/specificity.

The binding between FBXO42 and PP2A/PP6 is detected in the mass spectrometry presented in Figure 2A. Thus, we have now commented on the possibility that PP2A and

THE UNIVERSITY
of EDINBURGH

CANCER
RESEARCH
UK

Scotland
Centre

PP6 might be regulated as well (lines 481-483) but we feel that addressing properly the role of FBXO42 on all the PPP phosphatases would require a separate study given the complexity of their regulation. At the same time, the survival phenotype obtained after depletion of FBXO42 is fully rescued by PP4R2 depletion (Figure 5B), suggesting a prominent role of the FBXO42-PP4R2 axis in controlling the sensitivity to IR.

2) Is localisation of PP4 components affected by FBXO42 loss?

We measure the localisation of PP4C and PP4R2 in Appendix S2. In summary, we did not observe a significant difference in localisation of PP4C/PP4R2 comparing parental cells to cells lacking FBXO42.

3) Did the authors manage to map the ubiquitination sites on PP4C? Uniquely PP4C has two lysine residues at its very C-terminus (one has been reported to be ubiquitinated) and since the C-terminus of the PPP family is often a critical point of regulation it could be that these are the critical residues. Maybe this could be easily tested with K to R mutations.

Ubiquitylation of PP4C in the C-terminus has been reported (Phosphosite.org). However, mutating these residues did not impact the overall ubiquitylation of PP4C (Figure R1). In addition, the two lysine residues are in critical points of contact between PP4C and PP4R2-PP4R3A/B (Figure R2). Mutation of these lysine residues in PP4C would impair interaction of PP4C with PP4R2/PP4R3A,B by removing a critical hydrogen bond formed by the lysine residues (Figure R3). Although an attractive scenario is that FBXO42 ubiquitylates PP4C on the lysine residues and prevents interaction with PP4R2, mutations of lysine to arginine would prevent interaction with the other subunits, making the PP4C mutant an hypomorph, thus, not suitable in gain of function experiments.

4) An expansion of how FBXO42 recognises PP4 complexes and the role of CCDC6 in this would be interesting but not strictly needed.

The binding between FBXO42 and PP4 complex will likely require structural studies to clarify, something we are currently attempting. Simulations in AlphaFold of PP4C and FBXO42 interactions fail to predict a model with enough accuracy. FBXO42 kelch domain is unlikely to fold like other kelch proteins as the seventh blade of the β -propeller is separated by a large unstructured region. Although there is genetic evidence of a co-dependency between FBXO42 and CCDC6 (Depmap.org), the role of CCDC6 is complex and would require a separate study. Interestingly, CCDC6 phosphorylation is impacted significantly upon FBXO42 depletion with two phosphorylation sites being differentially regulated (Figure 4B).

5) Citations: please also cite Lipinszki et al 2015 as they were the first to describe the PP4 motif we just expanded on this work.

We apologise for this oversight; we have now referred the paper in lines 104 and 105.

Referee #2:

Major points

- Does FBXO42 knockout significantly alter the overall cell cycle distribution? This could be relevant to the interpretation phosphoproteomics and proteomics studies. A simple cell cycle analysis would be a helpful addition and could be done using the already generated KO cell lines.

There is a broad cell cycle disruption in cells lacking FBXO42 (Figure 1G), although the cells retain the capacity to replicate. Given the major role of FBXO42 in controlling PP4 (and probably other phosphatases), this is not surprising. Our interpretation is that the phosphorylation events controlling S phase checkpoint are disrupted inducing DNA damage and leading to G1 arrest and defective S phase. The proteomic and phosphoproteomic studies highlight a significant disruption of phosphorylation which may be due to PP4 phosphatase being more active and concomitant alteration of the cell cycle.

- Could some cellular phenotypes attributed to FBXO42 loss be rescued by depletion of PP4R2 or, maybe more difficult, PP4C?

We focused on the role of FBXO42 as a modulator of DNA damage response after IR. The sensitivity to IR in FBXO42 K/O can be fully rescued by depletion of PP4R2 (Figure 5B).

- Overall, the phosphorylation and proteomic analysis could be better described. The number of proteins and sites identified is unclear, as are the broad details for how the MS was done (e.g., DIA v TMT, fractionation, etc.). How was quantification performed? How many proteins or phosphorylation sites were identified?

We agree with the reviewer that we did not give enough weight to the description of the mass spectrometry study conducted. We have now more broadly described the MS approach in the text. We conducted the mass spec search in DIA mode and measured differences by free label quantitation. The phosphopeptide searches were conducted using ddaPASEF Data-Dependent Acquisition with the PArallel SEquential Fragmentation (PASEF), as described in the updated method section. We have now indicated both in the text and figure legends the cut-off used to score altered proteins and altered phosphopeptides.

- It is not clear what is being depicted in Figure 4A. Are these phosphorylation sites? If so, was only one site changed in each of these substrates? If it is a specific phosphorylation sites, what are those sites and were they the same sites identified in the previously done PP4 proteomics analysis? Relatedly, why do the authors refer to 14 proteins in the text, but show fewer in this figure?

We apologies for this oversight, we have now updated Figure 4A to reflect detected phosphopeptide alterations comparing parental cells to cells FBXO42 K/O. In the updated Figure 4B, we refer to the substrates of PP4R2 identified in Ueki et al., 2017, specifically the validated interactors highlighted in Figure 3C of the study. Please note Ueki et al. identified

interactors of PP4R2 and filtered them based on proteins containing FXXP and MXXP motifs. The phosphorylation sites directly regulated by PP4 are unknown. For this reason, we decided to run the search for phosphorylation of these proteins independently of where the phosphorylation sites reside. The proteins identified in Ueki et al., 2019 are 18 proteins. As indicated in the text of the 18 substrates, we did not detect 9, 2 had phosphorylation sites not changing, four had phosphorylation sites enriched in FBXO42 knockout and 6 had phosphorylation sites reduced in FBXO42 knockout. We used heatmaps to represent phosphopeptides altered (Figure 4B). Interestingly, CCDC6 had two phosphorylation sites being regulated in a diametrically opposite manner in FBXO42 knockout vs parental cells. Only the site closer to the C-terminus is reduced in FBXO42, this site is in proximity of the FXXP PP4R2 binding site. NCAPD2 phosphorylation site, reduced in FBXO42 knockout, has a canonical FXXP site within the peptide identified.

- What is the source of proteomics data analyzing PP4 knockdown by siRNA? Is this published, or is this a dataset generated here? Similar to the comment above, I believe this was done as part of this study, but the details of the experiment are not mentioned, number of sites identified, an experimental detail, etc. Importantly, how does this dataset overlap with the FBXO42 KO dataset (not just the putative kinases) and the previous publication on PP4 substrates?

We have generated siPP4 datasets in this study attempting to identify *bona fide* phosphorylation sites controlled by PP4 directly. We identified ~ 500 differential phosphopeptides (Figure EV4F), while there were no detectable differences in protein levels (Figure EV4D).

The overlap between PP4C known substrates and siPP4 is poor (Figure EV4G), an indication that this is not an approach suitable to identify PP4C substrates. Therefore, we comment in the text that we use this dataset as a proxy of altered PP4C activity in cells. The reason for this is likely due to the essential role of PP4 in controlling survival: we could only deplete ~50 % of PP4C (Figure EV4E) before starting cellular attrition.

In addition, we present in Appendix S3 the overlap of the FBXO42 K/O and siPP4C datasets. In summary, 45 phosphorylation sites were overlapping. 25 phosphopeptides are increased in siPP4C and are likely directly regulated by PP4 activity. Of these 25 sites, 10 are reduced in FBXO42 K/O settings.

- It would be helpful to show the data in Figure 4E is some sort of graph (bar graph?). As is, it is an unusual way to display these data, and it is not quickly apparent what it is.

We agree with the reviewer and have now substituted the table with a bar chart in Figure EV4D. In addition, we have complemented this dataset with immunoprecipitation of endogenous PP4C in FBXO42 knockout cells (Figure 5D).

Minor points

- BI2536 likely can inhibit other PLKs (mentioned in intro) We have now highlighted this in the introduction lines 138-140.

- The analysis in Figure 1 is compelling that FBXO42 is recurrently identified as being involved in the response to ATR inhibition and mitosis. It would be interesting to show an overlap from the strong hits of these screens. How many genes overlap between these

studies? Is FBXO42 the only E3 or F-box protein that scores in this analysis. A simple Venn diagram showing the overlapping hits would be interesting and providing these data in a supplemental table could be useful for others in the field interested in similar pathways. We would like to thank the reviewer for emphasizing this point as upon overlapping the different studies only FBXO42 remains a common hit (Venn diagram in new Figure 1D)

- Was the Enrichr analysis done on proteins that were altered by FBXO42 KO or on phosphoproteins altered? Similar to the comments above, the proteomics analysis and its discussion are in need of additional clarity throughout.

We now report the pathways corresponding to proteins depleted in FBXO42 K/O cells (Figure EV4B) and corresponding to proteins enriched in FBXO42 K/O cells (Figure EV4C). Distinct from the above, we also report the pathways enriched in the phosphorylations accumulating upon siRNA of PP4C (Figure EV4I) and the pathways enriched in the phosphorylations reduced in FBXO42 K/O (Figure EV4H). In both cases the top enriched pathways are mitotic spindle and G2-M checkpoint, an indication that similar processes are controlled in the indicated conditions.

- Clarify on line 327-328 that the upregulation referred to is phosphorylation. These lines have now been changed.

Referee #3:

Major Concerns

1. FBXO42 is reported to be a critical regulator of glioblastoma stem-like cells, as well as in melanoma and other cancer types. This study would benefit from further analysis of the effects of FBXO42-mediated PP4C regulation in cancer, using in vitro models, mouse models, and/or clinical tumor specimens.

We appreciate the comments highlighted here and agree that further studies are required to establish the role of FBXO42 in cancer models. However, these studies require dedicated models which take years to develop. We believe our study provides enough conceptual advances and mouse studies are currently out of scope.

2. The PP4C protein level is increased in FBXO42 KO cells, as shown in Figures 2D, 2E, and 4E. Is this due to post-transcriptional regulation or prolonged protein half-life? Although the authors excluded the possibility of increased protein stability, they should verify that their CHX assay was properly conducted by evaluating multiple cell lines and testing a range of CHX concentrations. A pulse-chase experiment is another widely accepted approach to confirm changes in protein stability and should be considered.

Two pulse-chase experiments are presented in Figures 2E (LN229 lines) and Figure EV2B (H4 lines). In both cases, it is possible to observe that the levels of the p53 protein are reduced following addition of CHX, a clear indication that CHX is working. Indeed, we reproduce the expected effect on mutant p53 reported in Lu et al, 2024; PMID:38580884

Minor Concerns

1. CCDC6 is presented as upregulated in both FBXO42-KO and parental cell groups in Figure 4A.

Please note these correspond to two distinct phosphorylation sites now presented in Figure 4B. The site identified being downregulated in FBXO42 is in proximity to a canonical FXXP binding site.

2. It would be beneficial to determine which lysine linkage is dominant in PP4C modification.

We are presenting now evidence suggesting that PP4C is modified by ubiquitin through K33 linkages (Figure EV3E and Figure EV3E). However, we are also wary that these chains could be extended due to non-physiological expression of the constructs. This is the reason why this finding is extensively discussed in lines 496 to 505.

Professor Vincenzo D'Angiolella, MD, PhD
Charles and Ethel Barr Chair of Cancer Research
The Institute of Genetics and Cancer
University of Edinburgh
Crewe Road South
Edinburgh
EH4 2XU

RF1

RF2

Lysine are labelled in green
 PP4C is labelled in Pink
 PP4R3A is labelled in orange
 PP4R2 is labelled in cyan

A

Flag

Ub

Overlay

4c

a tubulin

RF3

Selected lysine are labelled in red
 PP4C is labelled in Pink
 PP4R3A is labelled in orange
 PP4R2 is labelled in cyan

Reviewer Figure. Effect of selected K to R mutants in PP4C in LN229 and FBXO42 KO cells

1. Flag PP4C or the indicated K to R mutants were expressed in LN229 cells and FBXO42 KO cells. Flag-PP4C was isolated via Flag agarose beads pulldown before immunoblotting for ubiquitin.
2. Alpha fold model of PP4R2 complex with the all the lysine's in PP4C indicated in green.
3. Alpha fold model of PP4R2 complex with the selected lysine's in PP4C mutated to arginine indicated in red.

Prof. Vincenzo D'Angiolella
University of Edinburgh
Edinburgh Cancer Research Centre - Institute of Genetics and Cancer
2XU, Crewe Rd S, Edinburgh
Edinburgh
United Kingdom

28th Nov 2025

Re: EMBOJ-2025-121417R
Pervasive phenotypic effects of FBXO42 promoted by regulation of PP4 phosphatase

Dear Vincenzo,

Thank you for submitting your revised manuscript to The EMBO Journal. Two of the original referees have now reviewed it once more, and their comments are copied below. As you will see, they both appreciate your revisions, yet referee 1 still retains a few specific concerns, whose clarification would in my view be important to further strengthen the study prior to publication. I am therefore returning the manuscript to you for an additional round of revision, in which I would invite you to also incorporate the following formal/editorial issues:

- We still need you to complete all sections of the Author Checklist, including the general information table
- Please adjust the order of the manuscript sections, and also make sure to use the correct section headers: Title page with complete author information, Abstract, Keywords, Introduction, Results, Discussion, Methods, Data Availability, Acknowledgements, Disclosure and Competing Interests Statement, References, Main Figure Legends, Tables, Expanded Figure Legends.
- On the abstract page of the manuscript, please include 4-5 general keyword terms to enhance searchability. Furthermore, please remove the line called "Impact Statement" at the beginning of the manuscript.
- Please adjust the header and the format of the Disclosure and competing interests statement, and make sure to state if any author should be employed by a pharmaceutical/biotech company - for details, see <https://www.embopress.org/competing-interests>
- As we are switching from a free-text author contribution statement towards a more formal statement based on Contributor Role Taxonomy (CRediT) terms, please remove the present Author Contribution section and instead specify each author's contribution(s) directly in the Author Information page of our submission system during upload of the final manuscript. See <https://casrai.org/credit/> for more information.
- Please move the Reagents and Tools table from the main article file, and upload it as a separate text file. Also, please make sure to adhere to the template table downloadable from our author guidelines: <https://www.embopress.org/page/journal/14693178/authorguide#structuredmethods>
- Please make sure to adhere to our Author Guidelines regarding citation format throughout the whole text and the full reference list. Many entries are currently incomplete, lacking e.g. citation year, volume, or page/eLocator numbers. There should be no URLs in the reference list, and DOIs only for publications that do not have any other specific citation information yet. Also, please adjust the format for citation of preprints as specified in our author guidelines:
The citation in the text should be: "(preprint: et al.)"
The citation in the reference list: " , , ...
. bioRxiv doi:"
- Please double-check to make sure to all relevant funding information in the manuscript is congruent with the info entered into our submission system. Currently missing in the submission system are:
Glasgow Radiation Research Centre of Excellence RRCOER-Jun24/100003; John Fell Fund (133/075), Wellcome Trust (097813/Z/11/Z), and EPSRC (EP/N034295/1); the Chinese Academy of Medical Sciences (CAMS) Innovation Fund for Medical Science (CIFMS), China (grant number: 2018-I2M-2-002)
- Please double-check that each Figure panel is reference at least once in the text, e.g. Fig. 1H currently seems to be missing a call-out.
- In the Data Availability section, please remove the referee access information, and stick to the suggested wording: "The

[structural coordinates | microarray | mass spectrometry] data from this publication have been deposited to the [name of the database] database [URL] and assigned the identifier [accession | permalink | hashtag]."

Please also ensure that the data are becoming publicly available at this point.

- For the "supplementary tables", please turn them into Expanded View Datasets, and note that source file names, titles, legends and manuscript callouts all need to be updated to "Dataset EV1-EV4" instead of Supplementary Table 1-4. The respective legends should be included as a separate tab/sheet in each Excel file.

- Please pre-face the Appendix PDF with a title page stating "Appendix for " and a brief table of contents with the page numbers for the listed items. Please correct the nomenclature of the included figures to "Appendix Figure S1/2/3..." throughout the Appendix and the main text.

- During routine pre-acceptance checks, our data editors have raised the following queries regarding figures, data, and legends, which I would ask you to address (ideally using the Track Changes option):

1. Please note that the exact p values are not provided in the legends of figures 1E, 5B, C; EV1 G
2. Please indicate the statistical test used for data analysis in the legends of figures EV2 A, EV4 A, D, F
3. Please note that information related to n is missing in the legends of figures 5C, EV1 C, D, E, F; EV2 A, EV4 A, D, E, F
4. Please note that the error bars are not defined in the legends of figures 5C, EV1 C, D, E, F, EV4 E,
5. Please note that the measure of center for the error bars needs to be defined in the legends of figures 1E, 5B, EV1 G

- Finally, please provide suggestions for a short 'blurb' text prefacing and summing up the conceptual aspect of the study in two sentences (max. 250 characters), followed by 3-5 one-sentence 'bullet points' with brief factual statements of key results of the paper; they will form the basis of an editor-written 'Synopsis' accompanying the online version of the article. Please also upload a synopsis image, which can be used as a "visual title" for the synopsis section of your paper (maybe based on Fig 5E?). The image should be in PNG or JPG format, and please make sure that it remains in the modest dimensions of (exactly) 550 pixels wide and 300-600 pixels high.

I am therefore returning the manuscript to you for a final round of revision, to allow you to make these changes and upload all modified files. Should you have any questions regarding the referee comments or this decision, please do not hesitate to contact me directly.

With kind regards,

Hartmut

*** PLEASE NOTE: All revised manuscript are subject to initial checks for completeness and adherence to our formatting guidelines. Revisions may be returned to the authors and delayed in their editorial re-evaluation if they fail to comply to the following requirements. As a first step please read our guidelines for revised submissions:

<https://link.springer.com/journal/44318/submission-guidelines#cms-Revised-submissions>

1) Every manuscript requires a Data Availability section (even if only stating that no deposited datasets are included). Primary datasets or computer code produced in the current study have to be deposited in appropriate public repositories prior to resubmission, and reviewer access details provided in case that public access is not yet allowed.

4) Each main and each Expanded View (EV) figure should be uploaded as individual production-quality files (preferably in .eps, .tif, .jpg formats). For suggestions on figure preparation/layout, please refer to our Figure Preparation Guidelines:

<https://media.springernature.com/original/springer-cms/rest/v1/content/27825798/data/v1>

- 5) Point-by-point response letters should include the original referee comments in full together with your detailed responses to them (and to specific editor requests if applicable), and also be uploaded as editable (e.g., .docx) text files.
- 6) Please complete our Author Checklist, and make sure that information entered into the checklist is also reflected in the manuscript; the checklist will be available to readers as part of the Review Process File.
- 7) All authors listed as (co-)corresponding need to deposit, in their respective author profiles in our submission system, a unique ORCID identifier linked to their name. Please see our Guide to Authors for detailed instructions.
- 8) Please note that supplementary information at EMBO Press has been superseded by the 'Expanded View' for inclusion of additional figures, tables, movies or datasets; with up to five EV Figures being typeset and directly accessible in the HTML version of the article.
- 9) To facilitate reproducibility and cross-laboratory adoption of methodologies, please structure the Materials & Methods section as outlined in our guide to authors, including a completed Reagents and Tools Table.
- 10) Digital image enhancement is acceptable practice, as long as it accurately represents the original data and conforms to community standards. If a figure has been subjected to significant electronic manipulation, this must be clearly noted in the figure legend and/or the 'Materials and Methods' section. The editors reserve the right to request original versions of figures and the original images that were used to assemble the figure. Finally, we generally encourage uploading of numerical as well as gel/blot image source data.

In the interest of ensuring the conceptual advance provided by the work, we recommend submitting a revision within 3 months (26th Feb 2026). Please discuss the revision progress ahead of this time with the editor if you require more time to complete the revisions. Use the link below to submit your revision:

Link Not Available

Referee #1:

The authors have done a good job although I find the new activity assays difficult to evaluate as the input of PP4C is very low and impossible to evaluate if they IP similar amounts. It is in the PP4C IPs they see the biggest effect. Would be good with improved blots.

It is unclear from the MS analysis whether they have cell cycle differences in their different conditions which could lead to indirect effects. Please clarify.

I would use a different name of the PP4C-PPP4R2-PPP4R3A/B complex than PPP4R2 as it is the R3 subunit that binds FxxP motifs. Maybe R2/R3 complex to differentiate it from R1 complexes.

I would point out that FBXO42 is nuclear and thus has the same location as R2/R3 complex while CCDC6 is largely cytoplasmic and thus shares localisation with R1 complexes. Also I would point out that FBXO42 is almost 100 fold lower in concentration than PP4C arguing for a catalytic mechanisms of regulation while CCDC6 is more 1:1 with PP4:
Protein copy numbers from <https://pubmed.ncbi.nlm.nih.gov/28601559/>

CCDC6: 3,9 E+05
FBXO42: 4,8 E+03
PPP4R1: 7,9 E+04
PPP4R2: 1,7 E+05
PPP4C: 3,1 E+05
PPP4R4: 2,1 E+03
SMEK1/PPP4R3A: 9,4 E+04
SMEK2/PPP4R3B: 1,5 E+04

The layout of figures could need improvement and addition of MW markers

Referee #2:

The authors have addressed my prior concerns and I believe this manuscript is suitable for publication.

CANCER
RESEARCH
UK

Scotland
Centre

Professor Vincenzo D'Angiolella
Charles and Ethel Barr Chair Professor of Cancer Research
Edinburgh Cancer Research
Cancer Research UK Scotland Centre,
The Institute of Genetics and Cancer
University of Edinburgh
Crewe Road South
Edinburgh
EH4 2XU
Email: vdangio@ed.ac.uk

Dear Hartmut,

We have now reviewed our manuscript and implemented the corrections as suggested by you. Please also find below the answers to the comments of reviewer 1. We hope that the new updated version of the manuscript is a suitable candidate for *EMBO J*.

Referee #1:

The authors have done a good job although I find the new activity assays difficult to evaluate as the input of PP4C is very low and impossible to evaluate if they IP similar amounts. It is in the PP4C IPs they see the biggest effect. Would be good with improved blots.

The blots have now been changed in the updated Figure EV5C.

It is unclear from the MS analysis whether they have cell cycle differences in their different conditions which could lead to indirect effects. Please clarify.

We have highlighted that there are cell cycle differences in the cells FBXO42 K/O, presented in Figure 1G and Figure 1H. We can't exclude that the mass spec results also reflects cell cycle alterations.

I would use a different name of the PP4C-PPP4R2-PPP4R3A/B complex than PPP4R2 as it is the R3 subunit that binds FxxP motifs. Maybe R2/R3 complex to differentiate it from R1 complexes.

This has been changed as suggested.

I would point out that FBXO42 is nuclear and thus has the same location as R2/R3 complex while CDC6 is largely cytoplasmic and thus shares localisation with R1 complexes. Also I

THE UNIVERSITY
of EDINBURGH

CANCER
RESEARCH
UK

Scotland
Centre

would point out that FBXO42 is almost 100 fold lower in concentration than PP4C arguing for a catalytic mechanisms of regulation while CCDC6 is more 1:1 with PP4:
Protein copy numbers from <https://pubmed.ncbi.nlm.nih.gov/28601559/>

CCDC6: 3,9 E+05
FBXO42: 4,8 E+03
PPP4R1: 7,9 E+04
PPP4R2: 1,7 E+05
PPP4C: 3,1 E+05
PPP4R4: 2,1 E+03
SMEK1/PPP4R3A: 9,4 E+04
SMEK2/PPP4R3B: 1,5 E+04

We appreciate this observation and have discussed it in lines 485-488. We did not comment on CCDC6 as this is not the focus of our manuscript.

The layout of figures could need improvement and addition of MW markers

We have now added the MW markers in the uncropped images as adding the MW markers in the main figures would subtract from the clarity of the figures itself.

Please do not hesitate to get in touch for other information

Sincerely,

Professor Vincenzo D'Angiolella, MD, PhD
Charles and Ethel Barr Chair of Cancer Research
The Institute of Genetics and Cancer
University of Edinburgh
Crewe Road South
Edinburgh
EH4 2XU

Prof. Vincenzo D'Angiolella
University of Edinburgh
Edinburgh Cancer Research Centre - Institute of Genetics and Cancer
2XU, Crewe Rd S, Edinburgh
Edinburgh
United Kingdom

12th Dec 2025

Re: EMBOJ-2025-121417R1
Pervasive phenotypic effects of FBXO42 promoted by regulation of PP4 phosphatase

Dear Vincenzo,

Thank you for submitting your final revised manuscript for our consideration. I am pleased to inform you that we have now accepted it for publication in The EMBO Journal.

You may qualify for financial assistance for your publication charges - either via a Springer Nature fully open access agreement or an EMBO initiative. Check your eligibility: <https://link.springer.com/journal/44318/how-to-publish-with-us>

With kind regards,

Hartmut

Please note that it is The EMBO Journal policy for the transcript of the editorial process (containing referee reports and your response letters) to be published as an online supplement to each paper. If you should prefer removal of any referee-only figures included in the point-by-point response(s), e.g. because they may still be used for future publication or because they have been reproduced from published work by others, please do let us know immediately via response email.

More information is available here: <https://link.springer.com/partners/embo-press/editorial-policies#Peer%20review>